# CHARACTERIZING DEEP RESEARCH: A BENCHMARK AND FORMAL DEFINITION

**Abhinav Java**[*]
Microsoft Research
Bengaluru, India

**Ashmit Khandelwal**[*]
Microsoft Research
Bengaluru, India

**Sukruta Midigeshi**[*]
Microsoft Research
Bengaluru, India

**Aaron Halfaker**
Microsoft
Redmond, USA

**Amit Deshpande**
Microsoft Research
Bengaluru, India

**Navin Goyal**
Microsoft Research
Bengaluru, India

**Ankur Gupta**
Microsoft
Redmond, USA

**Nagarajan Natarajan**[†]
Microsoft Research
Bengaluru, India

**Amit Sharma**[†]
Microsoft Research
Bengaluru, India

## ABSTRACT

Information tasks such as writing surveys or analytical reports require complex search and reasoning, and have recently been grouped under the umbrella of *deep research* — a term also adopted by recent models targeting these capabilities. Despite growing interest, the scope of the deep research task remains underdefined and its distinction from other reasoning-intensive problems is poorly understood. In this paper, we propose a formal characterization of the deep research (DR) task and introduce a benchmark to evaluate the performance of DR systems. We argue that the core defining feature of deep research is not the production of lengthy report-style outputs, but rather the high fan-out over concepts required during the search process, i.e., broad and reasoning-intensive exploration. To enable objective evaluation, we define DR using an intermediate output representation that encodes key claims uncovered during search—separating the reasoning challenge from surface-level report generation. Based on this formulation, we propose a benchmark LIVEDRBENCH with 100 challenging tasks over scientific topics (e.g., datasets, materials discovery, prior art search) and public interest events (e.g., flight incidents, movie awards). Across state-of-the-art DR systems, F1 score ranges between 0.02 and 0.72 for any sub-category. OpenAI's model performs the best with an overall F1 score of 0.55. Analysis of the reasoning traces reveals that systems cover only about half of the necessary search queries, with proprietary models issuing broader and and deeper queries than open source models, highlighting gaps in both coverage and reasoning depth. The benchmark is available at `https://github.com/microsoft/LiveDRBench`.

## 1 INTRODUCTION

Deep Research (DR) has emerged as a popular paradigm for applying the reasoning capabilities of AI systems. DR systems have been shown to be useful for writing research reports (Li et al., 2025), business reports (Microsoft and Spataro, 2025), and answering needle-in-a-haystack queries (OpenAI, 2025). These tasks involve non-trivial reasoning that may take a human expert hours to complete and DR systems are expected to significantly reduce that time (FutureSearch et al., 2025; OpenAI, 2025).

Beyond demonstrative examples, however, no characterization of the problem exists. Typically, DR tasks are understood to be report-writing tasks that take non-trivial time for a human to complete. They are considered similar to reasoning-intensive search tasks, but expected to be much harder than

---

[*]Joint first authors, ordered alphabetically.
[†]Joint advising. Correspondence to: `nagarajn@microsoft.com, amshar@microsoft.com`

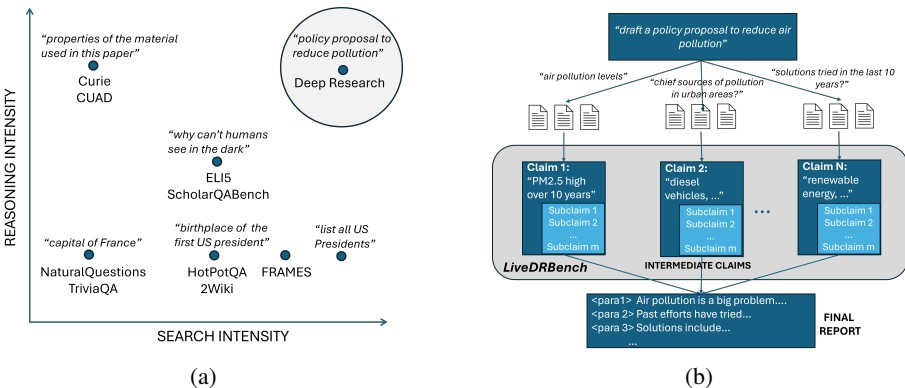

Figure 1: **Characterizing the deep research task.** Left (a) shows the landscape of various multi-hop reasoning tasks. Compared to existing tasks, the deep research task involves both high search and high reasoning intensity. Right (b) shows a stylized process of generating an answer to a DR query: DR query → Claims → Long form report. LIVEDRBENCH focuses on the precision and completeness of the intermediate but crucial step of claim generation.

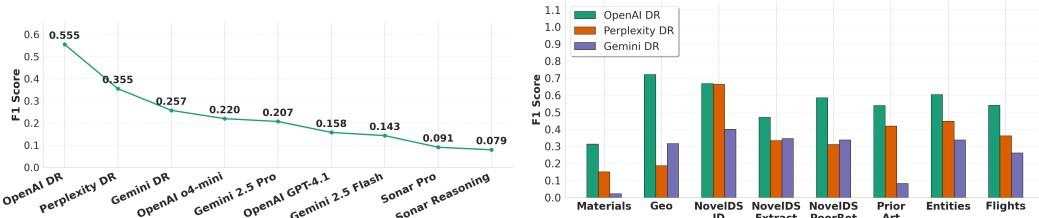

Figure 2: Average F1 score for deep research models and search-enabled LLMs on LIVEDR-BENCH.

Figure 3: Average F1 scores of deep research models split by the eight categories of LIVEDR-BENCH. OpenAI DR obtains the highest F1.

those studied in the literature such as multi-hop QA (Yang et al., 2018; Hongjin et al.). Interestingly, the hardness of a task also depends on the available document corpus. For example, "share a list of all Oscar-winning movies adapted from books with women authors" seems to be a hard, research-oriented task, but may not qualify as DR if there's a webpage providing exactly this information. Overall, the lack of a formal definition makes it hard to evaluate DR models and measure progress.

**Defining DR.** We characterize the deep research task and provide a benchmark that can evaluate progress objectively. Given a document corpus, we propose that the DR task can be split into two subtasks: 1) synthesizing *claims*—pieces of information relevant to a user's query, often supported by subclaims from retrieved evidence—and 2) writing a report based on these claims. Rather than the common expectation of a long report-like output, we posit that the *defining* challenge of DR is the first sub-task: synthesis of relevant information from a corpus. We conceptualize this as a directed acyclic graph (DAG), where the nodes represent information such as original query, retrieved documents and synthesized information, and edges represent actions such as issuing search queries and reasoning over retrieved documents (see Figure 1(b)).

We call a query a DR query if its information synthesis process is both search-intensive and reasoning-intensive. That is, given a document corpus, 1) the number of information units—atomic pieces of information such as a paragraph or a chunk—to be processed for obtaining the final answer is high (*search intensity*); and 2) at least one of the tasks—finding these information units, processing them or combining them to form the final claims—requires non-trivial reasoning from a human expert (*reasoning intensity*). Practically, we posit that any query that takes more than 10 minutes for a human expert qualifies as DR. Assuming the entire web as the retrieval corpus, examples include *"all oscar-winning movies adapted from books by women authors with runtime 80-95 minutes"* or *"find a material having the following properties"*. In contrast, examples of non-DR queries include "properties of the material used in this paper" (low search intensity) and "list of papers accepted at NeurIPS 2024" (low reasoning intensity).

| Benchmark | Objective Scoring | Multiple Claims in Response | Open Web Search | Easy to Update | Publicly Available |
|---|---|---|---|---|---|
| Deep Research Bench (FutureSearch et al.) | ✓ | ✓ | ✗ | ✗ | ✗ |
| BrowseComp (Wei et al.) | ✓ | ✗ | ✓ | ✗ | ✓ |
| DeepResearch Bench (Du et al.) | ✗ | ✓ | ✓ | ✗ | ✓ |
| DeepResearchGym (Coelho et al.) | ✗ | ✓ | ✗ | ✗ | ✓ |
| LiveDRBench | ✓ | ✓ | ✓ | ✓ | ✓ |

Table 1: Comparison of DR benchmarks. By employing a claim-based evaluation, LIVEDRBENCH can provide objective scoring for report-based DR answers that involve multiple claims. We focus on the open web search setup to allow direct measurement of the gap between proprietary and open DR systems, which necessitates mechanisms to easily update benchmark questions as the Web is updated.

The above definition distinguishes DR from commonly studied reasoning-intensive information tasks (Krishna et al., 2024; Yang et al., 2018; Ho et al., 2020; Hongjin et al.)(see Figure 1(a)). Multi-hop QA datasets like HotpotQA (Yang et al., 2018) are low in both search and reasoning intensity; scientific or legal QA tasks (e.g., CURIE (Cui et al.), CUAD (Hendrycks et al., 2021)) demand reasoning but little search; and explanation tasks such as ELI5 (Fan et al., 2019) involve reasoning but limited retrieval focused on a specific concept. In contrast, most DR tasks exhibit both high search and reasoning requirements, such as GAIA (Mialon et al., 2023) or Humanity's Last Exam (Phan et al., 2025). Beyond these *task completion* DR benchmarks that are evaluated on a single final answer, we focus on typical *information-synthesis* DR applications that involve a complex text output, such as writing a policy proposal, an academic survey, or an analytical report.

**Evaluating DR.** Given the complex output, assessing the quality of a detailed report is an inherently subjective task, involving evaluations of style and substance, as done by Du et al. (2025). Instead, for an objective evaluation, we evaluate a DR output only on the correctness and completeness of its substantive claims. Formally, we define the solution to a DR problem in terms of an intermediate output representation that consists of a nested list of claims: ⟨query, list of claims⟩ where an ideal solution gets all the claims and their (recursive) subclaims correct. Since users may typically evaluate both the claim and its supporting subclaims for assessing validity of the DR report, we define modified Precision and Recall metrics that assign zero credit if the claim or its subclaims are incorrect.

Based on this formulation, we provide a benchmark, LIVEDRBENCH, consisting of 100 DR queries spanning science and world events. In Table 1, we show the key benefits compared to other DR benchmarks (FutureSearch et al., 2025; Du et al., 2025): problem inversion technique (Wei et al., 2025a) for constructing DR queries that allows using the entire web as the retrieval corpus (easy comparison of both proprietary and open systems) and allows easy addition of new queries; objective evaluation based on our intermediate claim-based representation, and queries that capture utility of DR across society, including scientists, information professionals, and the general public. Specifically, the scientific domain contains queries to identify datasets or a material given certain properties and to identify whether a given idea is novel. The world events domain includes queries about flight crashes and entities such as books and movies. These queries span various types of DR queries, from needle-in-a-haystack to queries that require enumerating a list of entities.

We use LIVEDRBENCH to evaluate state-of-the-art DR systems from OpenAI, Perplexity AI, and Google, as well as the opensource Deep Researcher (Zheng et al., 2025) agent. LIVEDRBENCH presents challenging information-synthesis queries for these systems; the F1 score per category in these models ranges from 0.02–0.72 (see Figures 2 and 3). Overall, OpenAI's DR system obtains the highest (average) F1 of 0.55. Analysis of the reasoning traces reveals that systems cover only about half of the necessary search queries, with proprietary models issuing broader and and deeper queries than open source models. Our results motivate future directions on improving the search mechanisms for DR models.

## 2 BACKGROUND & RELATED WORK

We discuss recent deep research (DR) systems and their evaluation, as well as earlier efforts in long-form report generation which served as precursors. Our focus is on *information synthesis* tasks. While synthesis is core to DR, DR tasks may also involve computer use (Wei et al., 2025a), coding (Wang et al., 2025), or calling external tools (Mialon et al., 2023).

| Task | Description and Shortened Example | # Inst. |
|------|----------------------------------|---------|
| SCIFACTS Geo | Find research papers that use all datasets specified in the query. *Find the paper(s) that use all of the datasets mentioned: American Community Survey, Flooding Data from FEMA, ...* | 19 |
| SCIFACTS Materials | Find materials that match all specified measured properties. *Find the material(s) that satisfy every one of the measured properties: Direct band gap: 3.37 eV, Exciton binding energy: 60 meV, ...* | 17 |
| NOVELDS Identification | Find a dataset matching unique characteristics and return its metadata. *Identify a dataset of long-form in-the-wild videos that are segmented into scenes with ... and user engagement signals. The dataset should ... and support analysis at the video and scene level.* | 6 |
| NOVELDS Identification and Extraction | Find a dataset matching unique characteristics and extract specific findings from its paper. *I'm looking for a GPT-4 generated corpus of decisions rooted in quotidian life ... How does this corpus reveal GPT-4's implicit generation bias across various value dimensions ...?* | 11 |
| NOVELDS Peer Retrieval | Retrieve peer dataset papers in the same problem space based on a high-level description. *I want to compare existing 3D urban segmentation datasets ... that represent urban environments as richly annotated point clouds. How do they compare in terms of data acquisition methods such as ...?* | 3 |
| PRIORART | Find whether a proposed research idea has already been fully or partially explored in existing papers. *Can you help identify if the following idea has already been done ... in other papers? We develop a comprehensive evaluation framework for eliciting reasoning mistakes in LLMs ... explore aspects of mistake correction in LLMs ... and explicit mistake detection in reasoning chains ... We focus on distinguishing among self-generated responses ...* | 17 |
| ENTITIES | Find an exhaustive list of real-world entities matching detailed criteria. *Provide a comprehensive list films that meet the following criteria: 1. ... category of animation but not anime. 2. ... Tomatometer rating of 95% or higher. 3. .. runtime must be between 80 and 85 minutes inclusive.* | 20 |
| FLIGHTS | Find a real-world event that fits the described conditions. *In which flight incident did a commercial airliner perform an unusually high number of go-arounds before safely landing? For each attempt, detail the time, runway, landing aids ...* | 7 |
| **Total** | | **100** |

Table 2: Tasks in LIVEDRBENCH, spanning eight categories.

**Deep Research task.** In late 2024, Google launched a product for deep research queries (Google, b), followed by OpenAI's model in February 2025 (OpenAI, 2025), which was later integrated with Microsoft Copilot (Microsoft and Spataro, 2025). Perplexity (PerplexityAI, a) and Grok (xAI) DR systems soon followed, though details on design and evaluation of proprietary systems remain limited. Academic work on knowledge synthesis and long-form generation can be viewed as precursors to DR. Examples include scientific literature search (Asai et al., 2024), planning-based prompting (Godbole et al., 2024), and conversational prompting (Shao et al., 2024). After commercial launches, open-source implementations such as Huggingface Deep Research (Hugging-Face: Roucher et al., 2025), WebThinker (Li et al., 2025), Open Deep Research (Zhang, 2025), and DeerFlow (Bytedance, 2025) were proposed. These typically use either (i) agentic orchestration of LLM calls (HuggingFace: Roucher et al., 2025; Zhang, 2025; Bytedance, 2025) or (ii) reinforcement-learning finetuning (Shi et al., 2025; Li et al., 2025; Zheng et al., 2025). Although surveys on DR exist (Huang et al., 2025; Xu and Peng, 2025), a formal problem definition is lacking, which we provide in Section 3.

**Evaluating Deep Research.** Since DR output is typically a report, most evaluation focuses on long-form text quality (Du et al., 2025; Xu and Peng, 2025; Chandrahasan et al., 2025). DeepResearch Bench (Du et al., 2025) scores reports on comprehensiveness, insight, readability, and instruction following, using LLM judges such as GPT-4. ResearcherBench (Xu et al., 2025) uses human experts to create criteria for AI research queries, but is difficult to scale. In contrast, we provide an objective evaluation via claims, drawing on prior work in long-form evaluation (He et al., 2025) and claim verification (Metropolitansky and Larson, 2025; Dmonte et al., 2024). As for the query types, benchmarks either focus on needle-in-a-haystack queries Wei et al. (2025b) or focus on queries requiring long reports (Du et al., 2025; Xu et al., 2025). FutureSearch (FutureSearch et al., 2025) evaluates on economically valuable tasks (e.g., numeric answers, evidence gathering, claim validation), but its dataset is not public. Our approach differs by providing a public benchmark spanning both *needle-in-a-haystack* and broader *information-gathering* tasks (Table 2), with mechanisms to refresh questions to reduce contamination. Using the entire web as the corpus allows exact comparison of proprietary and open DR systems, unlike benchmarking efforts that build a static corpus (Kang and Xiong, 2024; Coelho et al., 2025). Furthermore, we formally define DR (Definition 1) to capture

both *search intensity* and *reasoning intensity*, and introduce a claims-based benchmark that measures both breadth (search coverage) and depth (claim correctness). These dimensions are not explicitly modeled in prior works like Mind2Web (Deng et al., 2023).

## 3 DEFINING DEEP RESEARCH

As discussed in Section 1, we focus on the DR subtask of generating the required claims to write a report. Writing a report given a set of claims is a long-form text generation problem (He et al., 2025) that can be independently studied.

**Defining an information unit.** The definition of deep research (and multi-hop RAG) depends on the retrieval corpus. Some queries may appear multi-hop (e.g., "movie adapted from the book that won the 40th Booker Prize") but are not because they can be answered from a single Wikipedia page (e.g., a paragraph in a page about the prize-winning book that also mentions the movie name). Other cases may involve combining information from multiple sections of a single large document (e.g., a 200-page financial report)—it is a single document but can be considered multi-hop due to multiple retrievals to fetch each section. Hence, we define an "information unit" to correspond to an *atomic* piece of information such as a paragraph, typically implemented as a chunk in retrieval systems (Liu et al., 2024). Consequently, a single document may include multiple information units.

### 3.1 DEEP RESEARCH: TASKS INVOLVING BOTH SEARCH AND REASONING INTENSITY

The deep research task can be considered as an *extreme* version of the multi-hop RAG task. Given a document corpus, answering a multi-hop RAG question requires retrieving and combining information from more than one document. There are three aspects in which the DR task is more complex than multi-hop RAG: how many pieces of information need to be processed; how easily those pieces of information are found, processed or combined; and how complex the final output is.

**1. Number of information units that need to be processed.** Unlike multi-hop tasks that are based on 3-5 information units (Yang et al., 2018; Ho et al., 2020; Krishna et al., 2024), many DR tasks require combining information from a large number of input sources, such as writing a survey based on tens of academic papers and content within them, or listing all items that meet a certain criteria. In terms of the search process of an ideal human expert, this corresponds roughly to the minimum number of searches issued by the expert.

**2. Complexity of finding, processing, or combining the required information units.** In many DR tasks, the search queries to issue are not obvious from the user's question and require tens of reasoning-intensive iterations. In other DR tasks, the search queries may be easy to enumerate, but processing or combining the retrieved information units to develop output claims is reasoning-intensive. Thus, a DR task includes at least one of the following reasoning components.

- *Finding information units.* For needle-in-a-haystack questions (e.g., "find material(s) with the following properties"), identifying the right search queries based on previous queries' results is the key subproblem.

- *Processing information units.* For other DR questions, the searches may be easy to enumerate, but processing each result may require reasoning. For example, consider the question, "English language films since 2024 meeting the following criteria...", where finding the listing of films is trivial but processing each film's properties needs reasoning.

- *Combining information units.* In other DR questions, the most difficult part is combining the retrieved information units to answer the user's question. For example, consider the question, "write a survey of open-weights language models released by American companies"—the most reasoning-intensive part is comparing information (e.g., latency) across results of multiple searches.

**3. Complexity of the Output.** Finally, a typical DR task requires a detailed report, involving multiple claims and their justification. We abstract out the content of the report as a nested list of claims, $\mathcal{A}$. Assuming independent top-level claims, the structure of $\mathcal{A}$ can be considered as a *list of dictionaries*, where each claim corresponds to a dictionary (see Figure 1b). Each top-level claim can have nested subclaims, which are the keys of the dictionary. We have the following definition for DR.

**Definition 1.** *Given a document corpus $\mathcal{C}$, query set $\mathcal{Q}$, and an ideal human expert who does not already know the answer to any query $q \in \mathcal{Q}$,*

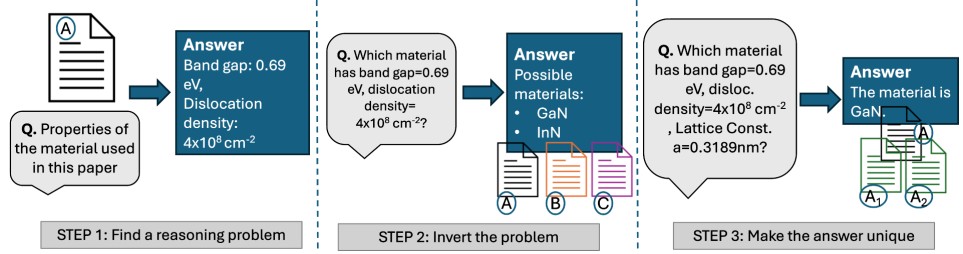

Figure 4: Problem inversion for LIVEDRBENCH. The first step is to find a long-context or document reasoning problem. In the second step, this problem is inverted to create a new question asking for an event or entity consistent with the properties mentioned in the document. In the third step, the question is refined (e.g., more properties are added) such that it admits a unique answer; and in case there are additional valid documents ($A_1$, $A_2$), the ground-truth set of documents is updated.

1. *A query q is said to be a deep research query if answering it requires processing a large number of information units (search intensity) and at least one of the following subtasks— finding the required information units, processing them, or combining them—requires non-trivial reasoning (reasoning intensity).*

2. *The deep research task can be formulated as a tuple $\langle$user query q, answer list $\mathcal{A}$, corpus $\mathcal{C}\rangle$ where $\mathcal{A}$ consists of independent claims that make up the final answer. Each claim in $\mathcal{A}$ may recursively include sub-claims that need to be derived and presented to arrive at the claim.*

In practice, *reasoning-intensive* for a human expert may translate to a solving time of $> 10$ minutes using tools such as web search, and search intensity to 20 information units (through $>= 10$ queries).

## 3.2 CLAIMS-BASED EVALUATION: PRECISION AND RECALL OVER GROUND-TRUTH CLAIMS

A key benefit of defining the DR output in terms of claims is that it unites the different kinds of DR tasks. Whether asking for a list of entities, a set of datasets or a particular material, each query's answer can be broken down into claims that are necessary to provide the answer, and in turn, sub-claims that are necessary to establish each claim. For developing an evaluation metric, we assume that the accuracy of subclaims is critical for the accuracy of a claim: 1) users may distrust claims that have incorrect subclaims, and 2) we want to reward DR systems that search a corpus and find the answer rather than those that rely on memorized knowledge.

From Definition 1, we assume that the ground truth can be written as $\mathcal{A}^* = [A_1^*, A_2^*, ...A_m^*]$ where each $A_i^*$ is a dictionary containing subclaims for the $i$th claim. Similarly, the DR output can be parsed into a list of claims, $\mathcal{A}$ (for an example prompt, see Appendix F). Let $s_{A_i}$ be the agreement score between any claim $A_i$ and the ground truth claim $A_i^*$. Typically, this will be a binary score $\in \{0, 1\}$. For each independent claim, its contribution to the Precision metric is the product of its own agreement score and the precision of its subclaims (which would be an average over agreement scores for all the subclaims). For the recall metric, we follow the same approach.

$$\text{Prec}(\mathcal{A}) = \frac{\sum_{A_i} w_i s(A_i) \text{Prec}(A_i)}{\sum_{A_i} 1}; \ \ \text{Rec}(\mathcal{A}) = \frac{\sum_{A_i} w_i s(A_i) \text{Rec}(A_i)}{\sum_{A_i^*} 1} \quad (1)$$

where $\text{Prec}(A_j) = \text{Rec}(A_i) = 1$ whenever $|\text{subc}(A_i)| = 0$. Given a claim $A_i$, $\text{subc}(A_i)$ represents its subclaims; $\text{subc}(A_i) = \phi$ for an atomic claim. $w_i$ is an optional weight that can be assigned to each claim, e.g., to weight the most important claims higher. Note that we can also have a *strict* version of metrics that provide a score of zero if any of the subclaims is incorrect (see Appendix B).

## 4 LIVEDRBENCH: A CHALLENGING BENCHMARK FOR OPEN WEB DR

We start by discussing the desiderata for a DR benchmark, present the problem inversion technique that helps us create LIVEDRBENCH satisfying the desiderata, and then give details on the tasks.

**Criteria for tasks.** Formulating Deep Research (DR) evaluation tasks is tricky because the models typically have access to the entire Web at inference time. We list the following desiderata for a DR

evaluation benchmark:

**(1)** The tasks should not be answerable using any single Web source. This is especially challenging, as the Web is constantly updated and a new article specific to a query may collapse a DR task into single retrieval one.

**(2)** The tasks should require accessing a wide variety of sources on the Web, to force the models to find the right grounding rather than relying on parametric memory.

**(3)** Evaluation of each task should be objective and reproducible, and metrics should help benchmark the progress in the DR field.

**(4)** The tasks should be amenable to periodic updates. This helps keep pace with fast-moving Web as well as fast-changing models.

**(5)** The tasks should capture a wide variety of DR domains and users (e.g., scientists, researchers and analysts, common Web users).

Striking a balance between all these constraints is non-trivial. For instance, constructing a benchmark by asking domain experts to provide the questions, as by Du et al. (2025), satisfies the second property, but it may not be easy to update. Instead, we leverage a strategy based on *inversion* of standard NLP and reasoning problems (see Figure 4), described next.

**Problem inversion technique.** We build on the insight used in the construction of OpenAI's recent benchmark (Wei et al., 2025a)—craft questions for which the answers are hard to *find* but easy to *verify*. However, their procedure requires asking human raters to think of a concept or fact with many unique characteristics and *invert* it to ask a question to find such a concept. To craft DR questions in a scalable way, we *templatize* the procedure and apply it to different kinds of information sources such as research publications and publicly available reports or lists. As publications or public reports keep updating, we can easily update our benchmark questions. More importantly, our construction is flexible and allows creation of questions whose answers include multiple entities and claims.

As shown in Figure 4, the proposed inversion process involves three steps: **1)** identify documents–typically long documents—that include detailed information; **2)** extract entities or concept classes described therein with unique characteristics; and **3)** devise a DR question to find such entities having the said characteristics, without knowledge of the input documents. In particular, long-context reasoning problems are especially suitable for inversion, since they already include questions that can be used to identify the entities. The standard long-context reasoning problem has the following flavor, as in the Curie benchmark for reasoning over scientific papers (Cui et al.):

[Long-context Reasoning] *Given a long source document on an entity or a subject and a list of keys, identify values for the keys consistent with the document.*

For instance, given a research article about the effects of a certain treatment, answering questions about the subjects of study, statistical significance, etc. would fall under long-context reasoning. The inverted problem becomes a DR task:

[DR] *Given keys and values that conform to a certain (unique) entity, identify the entity and locate the sources that corroborate the identification.*

In the above example, the inverted DR task is to locate the research study article(s) given the parameters of user study, participant details, symptoms, and chemicals used in the treatment. Note that although we started with a single research article, it is possible that the dictionary of ⟨keys, values⟩ may yield multiple articles as the correct answer. In practice, we manually verify that a DR question yields a unique answer and if not, we update the extracted dictionary (or in some cases, extend our ground-truth manually by adding the additional correct answers). In other cases, multiple entities form the answer by design. For example, consider the (hypothesized) long document combining all Oscar award announcements till date; we can ask inverted questions about any subset of awardees that satisfy certain relevant criteria.

**LIVEDRBENCH tasks.** Based on the aforementioned desiderata (1–5), our tasks cover the domains of science (facts and datasets), innovation (prior art search), and global culture and events (entities and flights). As discussed in the Introduction, we formulate tasks that expect claims with right grounding as output (see Figure 1) and ensure that these tasks have a range of reasoning and search requirements. A majority of tasks in LIVEDRBENCH are based on inverting the long-context reasoning problem (as discussed above). The eight task categories in LIVEDRBENCH are listed in Table 2. We give details of task construction in Appendix C and examples in Appendix D. Below, we give a brief description.

**(1-2) SCIFACTS.** We create 31 questions, derived from scientific papers and their questions from the

CURIE dataset (Cui et al.), to cater to scientists and researchers: (1) the **Materials** subset consists of 17 questions emulating scenarios where a scientist might want to look for a specific material that perfectly fits the given criteria. The ground-truth answer for each question consists of the material name (*claim*) and the source paper title (*subclaim*); (2) the **Geo** subset consists of 19 questions emulating a researcher attempting to search for papers that precisely utilize a given set of geospatial analysis datasets. The ground-truth answer is a list of paper titles (*claims*) which use given datasets. **(3-5) NOVELDS.** We create 20 questions based on dataset papers, drawing from both our domain knowledge and the dataset papers accepted at ICLR 2025: (3) the **Dataset Identification** questions are designed to emulate scenarios where a researcher is looking for details of a dataset satisfying specific characteristics. The ground-truth answers consist of dataset name (*claim*) and meta-data such as publication yer, venue, and source link (*subclaims*); (4) **Dataset Identification and Extraction** questions also require extracting specific results (e.g., interpreting a figure) from the dataset paper (*subclaims*); (5) the **Peer Dataset Retrieval** questions elicit articulating the key features that define and distinguish multiple related datasets pertaining to a given high-level description. The ground-truth answers include dataset details (*claims*) and supporting details such as performance metrics or dataset attributes (*subclaims*).

**(6) PRIORART.** We devise 17 questions motivated by the laborious problem of validating (or invalidating) claims in patent applications (Churnet, 2012). Using recent ML papers from ICLR 2025 and ICML 2025, we manually combine ideas and contributions from 2-3 papers to write a new paper abstract, and the DR question is to identify the key ideas and the source papers that demonstrate those ideas. The ground-truth answer consists of a list of ideas or contributions in the abstract (*claims*), paper titles and links of the source papers and rationales (*subclaims*) for why the references demonstrate the claims.

**(7) FLIGHTS.** We construct 7 questions based on official flight investigation reports published by national aviation authorities. Each question involves inverting the report: identifying the correct flight incident and specific details, given a high-level description elaborating the causes or timeline of the incident. The ground-truth answer is identification details such as timestamp (*claims*) and supporting details such as sensor readings (*subclaims*).

**(8) ENTITIES.** We devise 20 questions pertaining to global culture (e.g., sports, books) that require consulting a broad range of sources on the Web. These questions pertain to lists of entities connected to an event (e.g., a yearly award like Oscars). They are constructed by inverting the set of documents corresponding to a global event or entity: developing a certain criteria and asking a question to produce an exhaustive list of entities satisfying the specified criteria. The ground-truth answer is a list of entity names (*claims*).

**Updating the benchmark.** As remarked above, our templatized inversion process makes it easy to update the benchmark as new scientific papers or global events happen. SCIFACTS can be augmented with other long-context reasoning benchmarks such as DocFinQA Reddy et al. (2024),NOVELDS with new dataset papers, FLIGHTS with new flight incidents, PRIORART with new ML papers, and ENTITIES with new editions of the global events such as awards.

## 5  USING LIVEDRBENCH TO EVALUATE DEEP RESEARCH SYSTEMS

**Setup.** We evaluate three proprietary DR models: OpenAI DR (OpenAI, 2025), Perplexity DR (PerplexityAI, a), and Gemini DR (with 2.5 Pro) (Google, b). Among open DR models, we consider DeepResearcher (Zheng et al., 2025); others such as HuggingFace DR (HuggingFace: Roucher et al., 2025) and LangChain Open DR (LangChain, 2025) did not run successfully due to infinite looping or system errors. In addition, we also evaluate three reasoning-enabled baselines: OpenAI's o4-mini (OpenAI, b), Perplexity's Sonar Reasoning (PerplexityAI, c), and Gemini 2.5 Pro (Google, a), and 3 non-reasoning baselines: OpenAI's GPT-4.1 (OpenAI, a), Perplexity's Sonar Pro (PerplexityAI, b), and Gemini 2.5 Flash (Google, a). All baselines have *web search* enabled. Additional setup details are discussed in Appendix A, and additional evaluation results are discussed in Appendix I.

### 5.1  RESULTS

**Overall.** Table 3 reports precision, recall, and F1 scores for all Deep Research models across all task categories. The claim agreement scores are computed using GPT-4o (see Appendix C.3 for details) and the metrics are computed using Equation 1. Across all categories, the OpenAI DR model

| Subset | OpenAI Deep Research | | | Perplexity Deep Research | | | Gemini Deep Research (2.5 Pro) | | | DeepResearcher + DS Qwen 32B | | |
|---|---|---|---|---|---|---|---|---|---|---|---|---|
| | Prec | Rec | F1 | Prec | Rec | F1 | Prec | Rec | F1 | Prec | Rec | F1 |
| SCIFACTS Materials | 0.313 | 0.316 | **0.314** | 0.264 | 0.105 | 0.150 | 0.072 | 0.013 | 0.022 | 0.000 | 0.000 | 0.000 |
| SCIFACTS Geo | 0.715 | 0.728 | **0.721** | 0.263 | 0.144 | 0.186 | 0.405 | 0.259 | 0.316 | 0.000 | 0.000 | 0.000 |
| NOVELDS Identification | 0.667 | 0.667 | **0.667** | 0.633 | 0.633 | 0.633 | 0.400 | 0.400 | 0.400 | 0.167 | 0.167 | 0.167 |
| NOVELDS Identification and Extraction | 0.526 | 0.448 | **0.470** | 0.325 | 0.349 | 0.333 | 0.406 | 0.329 | 0.345 | 0.045 | 0.015 | 0.023 |
| NOVELDS Peer Retrieval | 0.795 | 0.480 | **0.585** | 0.494 | 0.248 | 0.311 | 0.551 | 0.249 | 0.338 | 0.083 | 0.028 | 0.042 |
| PRIORART | 0.694 | 0.463 | **0.539** | 0.689 | 0.339 | 0.419 | 0.106 | 0.096 | 0.082 | 0.525 | 0.136 | 0.199 |
| ENTITIES | 0.813 | 0.534 | **0.603** | 0.673 | 0.373 | 0.447 | 0.494 | 0.3 | 0.338 | 0.307 | 0.046 | 0.076 |
| FLIGHTS | 0.542 | 0.546 | **0.540** | 0.389 | 0.347 | 0.362 | 0.249 | 0.276 | 0.261 | 0.120 | 0.073 | 0.090 |
| **Average** | 0.633 | 0.523 | **0.555** | 0.466 | 0.317 | 0.355 | 0.335 | 0.240 | 0.263 | 0.156 | 0.058 | 0.075 |

Table 3: Precision, recall, and F1 scores on LIVEDRBENCH for Deep Research models.

| Model | Precision | Recall | F1 |
|---|---|---|---|
| OpenAI DR | 0.526 | 0.448 | **0.470** |
| Perplexity DR | 0.325 | 0.349 | 0.333 |
| Gemini DR | 0.406 | 0.329 | 0.345 |
| DeepResearcher + DS Qwen 32B | 0.045 | 0.015 | 0.023 |
| DeepResearcher + GPT-4.1 | 0.167 | 0.090 | 0.101 |
| OpenAI o4-mini | 0.203 | 0.146 | **0.168** |
| Sonar Reasoning | 0.015 | 0.003 | 0.005 |
| Gemini 2.5 Pro | 0.186 | 0.130 | 0.142 |
| OpenAI GPT-4.1 | 0.126 | 0.078 | 0.088 |
| Sonar Pro | 0.027 | 0.020 | 0.023 |
| Gemini 2.5 Flash | 0.211 | 0.097 | **0.111** |

Table 4: Comparison of DR models and baselines on the NOVELDS Identification and Extraction.

| DR Model | % Nec. Coverage | # Dep. Queries | # Indep. Queries |
|---|---|---|---|
| OpenAI DR | 66.0 | 34 | 64 |
| Perplexity DR | 52.0 | 15 | 25 |
| Gemini DR | 46.8 | 39 | 24 |
| DeepResearcher + DS Qwen 32B | 53.4 | 5 | 5 |
| DeepResearcher + GPT-4.1 | 49.4 | 5 | 6 |

Table 5: Comparison of DR models by Necessary Query Coverage and Dependent/Indepedent queries.

achieves the best performance, often significantly better than Perplexity and Gemini DR models, with an overall average F1 of 0.55. While Gemini outperforms Perplexity on SCIFACTS Geo, NOVELDS Identification and Extraction, and Peer Retreival, Perplexity exhibits stronger average performance compared to Gemini. Across models, the F1 score per category ranges from 0.0 (DeepResearcher on SCIFACTS Materials) to 0.72 (OpenAI DR on SCIFACTS Geo). The open-source DeepResearcher model obtains significantly lower scores (except for PRIORART where it outperforms Gemini DR), highlighting a big gap in proprietary and open DR systems. These results also highlight the difficulty of LiveDRBench compared to other benchmarks. In particular, non-reasoning models such as GPT-4.1 achieve low F1 scores (0.0-0.33 across categories, e.g., see Table 4). Category-wise results that include non-DR models are in Tables 7-13 in Appendix.

**Accuracy of claim and its subclaim.** The DR models perform worst on SCIFACTS Materials, NOVELDS Identification and Extraction, and FLIGHTS, likely because these tasks require reasoning across multiple sections of papers and flight reports to extract both main and subclaims. The models perform best on SCIFACTS Geo, which does not involve any subclaim extraction. Similarly, the performance NOVELDS Identification is higher due to its relative simplicity – paper, venue, and its link are closely related. For SCIFACTS materials, models often correctly extract either the paper title or the material name, but not both, resulting in low overall F1 scores. OpenAI DR achieves an F1 of 0.735 for paper titles and 0.504 for materials, but only 0.314 overall. Similarly, PPL DR reaches 0.348 (title), 0.32 (material), and 0.158 (overall), while Gemini DR performs worst with 0.17, 0.305, and 0.023, respectively. This highlights the difficulty of accurate extraction of claims.

## 5.2 Analysis of Deep Research Models

We analyze the search queries present in the DR models' reasoning traces to quantify **(1)** the coverage over necessary search queries (as a proxy for breadth of search), **(2)** their dependence on preceding queries (as a proxy for depth); **(3)** amount of branching and backtracking employed. GPT-4o is used to analyze the reasoning traces (prompts in Appendix G).

**Coverage over necessary search queries.** To evaluate whether the DR models invoke the complete set of required queries for a given DR question, we generate search queries without which it is not possible to fully answer the question. Specifically, we generate `characteristic queries`, which are queries that need to be issued to locate relevant source papers or reports, and `extractive queries`, which target the fields to be retrieved from those sources. Coverage is then measured as a percentage of these necessary queries that are present in the DR models' traces. Results in Table 5 show that DR systems can generate nearly half of the required queries, demonstrating a big potential for improvement. Qualitative examples are included in Appendix H.

**Dependence on preceding queries.** To approximately assess the depth of the DR models' traces, we analyze the dependence of each query on preceding queries. Using an LLM prompt, each query in the reasoning trace is classified as either `Independent`, if it initiates a new line of inquiry, or `Dependent`, if it builds upon earlier queries. Results highlight the difference in capabilities between open DR and proprietary models. Table 5 shows that proprietary models issue significantly more queries (24-64 on average) than DeepResearcher (5-6), which also leads to more dependent queries (15-39 on average), going deeper over many topics, than DeepResearcher (5).

**Branching and Backtracking.** Both these operations are measured using GPT-4o (see Fig. .5). Number of branches can be considered as another aspect of breadth (beyond necessary queries); while backtracking denotes robustness of reasoning to incorrect reasoning paths. OpenAI DR has the highest number of branches, supporting its high necessary queries' coverage score; and suggesting the key role of breadth in DR answer accuracy. We do not see a significant difference in backtracking.

## 6 Discussion and Conclusion

Our evaluation points to a few directions for improving DR. DR models do not perform well even when the algorithm is straightforward but the search may be laborious (e.g., Entities); interleaving programmatic and model control may be helpful for such tasks. For other problems such as SciFacts, DR models would often yield the correct answer but wrong grounding, which may be addressed by better training. Finally, our trace analysis can help guide tradeoffs of breadth and depth in DR.

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

# A  DETAILED SETUP

At the time of writing this paper, the proprietary DR models did not have API access. So we evaluate these models in their chat interface by manually feeding in the DR tasks in our benchmark. Typically, DR models in these chat applications respond with clarification questions in the beginning. We let the models use their best judgment with a fixed response to the effect of "Go Ahead".

For the baselines, we rely on the APIs provided by each model's provider. All hyperparameters are kept at their default settings, except for the search context size and reasoning effort/budget parameters. For search context size, we chose "`medium`" for OpenAI and Perplexity baselines, with the Gemini API not providing an equivalent parameter. For reasoning effort/budget, we chose "`low`" for o4-mini, "`medium`" for Sonar Research, and `128 tokens` for Gemini 2.5 Pro – each corresponding to $\approx$30 seconds of thinking time by the respective models.

# B  EVALUATION: STRICT PRECISION AND RECALL

The strict version of the score takes the minimum instead of the mean of the constituent claims' scores. This formulation is motivated by user trust in the final report: a user is unlikely to trust (even a correct) claim if any of its derivatory subclaims are incorrect.

$$\text{Prec}(\mathcal{A}) = \min_{i \in \{1,2,\dots|\mathcal{A}|\}} w_i s(A_i) \, \text{Prec}(A_i); \ \ \text{Rec}(\mathcal{A}) = \min_{i \in \{1,2,\dots|\mathcal{A}^*|\}} w_i s(A_i) \, \text{Rec}(A_i)$$

where $\text{Prec}(A_j) = \text{Rec}(A_j) = 1$ when $|\text{subc}(A_j)| = 0$ as before. Effectively, such a score will be 1 only if the DR model's output exactly matches the list of dictionaries $\mathcal{A}$: no claim is incorrect and no claim is missing. Such an evaluation may be useful when asking for an enumeration of a certain class of entities, e.g., all movies that satisfy a constraint. One may consider a DR system's output a failure if it fails to retrieve exactly the number of movies that satisfy the constraint.

# C  TASK CATEGORIES, UPDATION AND EVALUATION

## C.1  TASK CATEGORIES

### C.1.1  SCIFACTS: SOURCING SCIENTIFIC FACTS

We create 31 queries designed to cater to scientists and researchers. This dataset has two distinct subsets - (a) material identification, and (b) geospatial paper identification. Below we discuss both these categories in detail.

**Materials.** We create **17 questions** to emulate scenarios where a scientist might want to look for a specific *compound or material* that perfectly fits a given set of criteria. Each question comprises a list of measured properties that maps perfectly to a compound; and the task of a deep research agent is to successfully identify this compound, along with the source of information (academic paper). We leverage the *mpve* subset from the CURIE (Cui et al.) dataset to construct a valid set of ground truth answers. The CURIE (Cui et al.) dataset consists of long-form comprehension and reasoning questions. Here, a subset of questions require the model to enumerate all the measured properties of a given material from a target paper. To convert this to a deep research query we leverage the ground truth measured properties and design the task such that the job of the model is to identify the material, reference paper title, inference basis, and extract the relevant tables and passages from the reference paper. We compute final metrics based on the material name and paper title.

**Geo.** We create **19 questions** for paper identification to mirror the scenario of a researcher attempting to search for citations that precisely utilize a given set of *geospatial analysis datasets*. The task of the deep research agent is to find papers that utilize all the datasets listed in the question. Similar to "Material Identification" (above), we utilize CURIE (Cui et al.) to construct our ground truth corpus from their *geo* subset consisting of papers from the geospatial analysis field. The *geo* task in CURIE involves listing all datasets used in a given research paper, and we *invert* this task to construct a DR question to find the paper.

**Ground-Truth Claims.** In the SCIFACTS Geo task, the ground truth is provided as a list of paper titles which use the datasets mentioned in the original query. Similarly, in SCIFACTS Materials task, we provide a list of material names (*claims*) and associated supporting evidence such as paper titles (*subclaims*) as the ground truth and compute the metrics according to Eq. 1. Since the material identification task consists of both the main claim and the sub-claim, it is significantly harder than the geo task.

### C.1.2  NOVELDS: IDENTIFYING DATASET PAPERS

We create 20 questions based on dataset papers, drawing from both our own domain knowledge and the dataset papers accepted at ICLR 2025. This category contains three types of tasks, (a) Dataset Identification, (b) Dataset Identification and Extraction, and (c) Dataset Peer Retrieval. Unlike the Geo category from SCIFACTS, here the goal is to output dataset(s) and their associated properties.

**Dataset Identification.** These questions are designed to emulate scenarios involving a researcher looking for a dataset for a specific use-case. For each dataset paper, we identify its core contributions and the aspects that distinguish it from related work, to create open-ended queries that a human might realistically pose when unaware of the dataset's existence but seeking something with those specific characteristics. The goal is to reflect a genuine information need, rather than merely paraphrasing the abstract. The task requires the model not only to identify the appropriate dataset, but also to extract basic metadata such as its publication year, venue, and a link to the dataset. Notably, failing to account for even one of the dataset's unique attributes often results in the retrieval of a similar yet ill-fitting alternative.

**Dataset Identification and Extraction.** These questions build upon the Dataset Identification category by requiring not only the identification of the correct dataset but also the extraction of specific findings from the dataset's paper. These questions mimic those a researcher might pose when seeking results or findings for a specific use case, without being aware of the dataset's existence. Extracting these findings may, for example, involve reasoning over results in a table, reasoning over multiple tables, or interpreting a figure in a paper.

**Peer Dataset Retrieval.** Some dataset papers include literature surveys or comparison tables highlighting peer datasets with similar goals, and slight variations in task settings. We use these sections to construct questions that reflect shared research directions across multiple datasets, without referencing any dataset by name. These questions emulate real-world exploratory scenarios in which a researcher is familiar with the broader problem space but not with specific datasets. The task requires the model to search and reason given a high-level description to find a set of candidate datasets, and then to articulate the key features that define and distinguish each one.

**Ground-Truth Claims.** For the NOVELDS Identification task, we provide the ground-truth paper title, publication year, venue, and dataset link required by the queries. Here, paper title serves as the main claim, while the year, venue, and link form the subclaims. For NOVELDS Identification and Extraction and NOVELDS Peer Retrieval, we provide a list of ground-truth findings or peer datasets required by each query. Each item in the list includes a set of main claims (e.g., "task", "model-name", "dataset-name"), with supporting details such as performance metrics or dataset attributes serving as subclaims. Precision, recall, and F1 scores are computed based on the overlap of the model outputs with that of the ground-truth.

### C.1.3  PRIORART: DECONSTRUCTING PAPER ABSTRACTS

We devise 17 novel tasks motivated by the laborious problem of validating (or invalidating) claims in patent applications Churnet (2012). Typical response to a patent application consists of a list of claims that are accepted and those that are rejected. For each rejected claim, the response also includes citations that are prior art that already demonstrate the claims in whole or in parts, thereby challenging the grounds for patenting. To mirror this real-world scenario, we create DR tasks that involve deep reasoning over synthetic research paper abstracts.

The task is to reason over the given paper abstracts, manually written by us to mimic the paper abstracts in standard ML venues. Each abstract *combines* ideas and contributions primarily from recent ML papers in ICLR 2025 and ICML 2025 conferences (including accepted and rejected papers). The goal is to understand the key ideas in a given abstract and identify papers that were the

source of those ideas, i.e., that directly or indirectly provide evidence to support the ideas. Each task requires the model to list the claims in the abstract, references (paper titles and links), and rationales for why the references demonstrate the claims.

**Dataset creation.** We rely on OpenReview discussions to create the tasks. We start by identifying papers in ICML 2025 and ICLR 2025 venues on OpenReview where there is sufficient engagement from the reviewers and the authors during the rebuttal phase. For each of the identified papers, we start with the paraphrasing of the contributions from the meta-reviewer, reviewers, and in some cases, the authors themselves. This becomes the seed for the synthetic abstract in the task. We find that this seed is already challenging to identify, as the paraphrasing is often nuanced, highlighting aspects beyond the corresponding published paper abstract (in particular, lexical matching of the seed text with the original paper abstract wouldn't be helpful). We then go through the discussions to identify closely related papers pointed out by the reviewers. We use the discussions as well as the contributions stated in the text of the related papers to augment the synthetic abstract for the task.

For instance, if the seed text is about a method that works under the setting of Scenario A, and the reviewers point out a closely related paper that solves the same problem in Scenario B but using a modified method, then we write the synthetic abstract to subsume both the scenarios as well as the modifications of the method as purported contributions.

In most cases, the related work pointed out by the reviewers are papers that *are not already cited by the paper at hand*. Furthermore, in some cases, we found a few other papers that also are closely related to some the claims in the abstract. In those cases, we manually added those references to the ground-truth. Overall, this forms a challenging DR problem — the models cannot simply follow the references in the related work of one paper to get all the source papers for the task. In this way, our tasks that require deconstructing paper abstracts are more nuanced than standard DR tasks that elicit literature review or drafting related work given a topic.

**Ground-Truth Claims.** The ground-truth consists of top-level claims, where each claim corresponds to an idea or a contribution discussed in the abstract. Each claim is supported by the sub-claims consisting of ground-truth paper title, link, and a connection field that describes how the claim is demonstrated in the referenced paper. The DR model is expected to produce the same format in its output. We compute precision, recall, and F1 scores based on the overlap of the model-reported references with that of the ground-truth.

### C.1.4   FLIGHTS: IDENTIFYING FLIGHT INCIDENTS

**Dataset Creation:** We construct 7 questions based on official flight investigation reports published by national aviation authorities (e.g., the NTSB, AAIB, and BEA). Each question is derived by identifying a flight incident with unique features – such as an unusual failure mode, a rare technical error, or a non-standard response procedure. We then *invert* the task: the DR question involves identifying the correct flight incident from a high-level description, elaborating on its causes or timeline.

These questions emulate scenarios in which members of the general public, aviation enthusiasts, or journalists seek information about notable incidents without knowing the specific flight number. As official incident reports often span hundreds of pages, the task requires processing long documents and synthesizing information across multiple sections.

**Ground-Truth Claims.** We provide a ground-truth list of the incidents' timelines or causes, as required by each query. Each item in the list includes a set of main claims (e.g., "timestamp", "flight-parameter"), with supporting details such as sensor readings, flight configurations, anomaly descriptions serving as subclaims. Precision, recall, and F1 scores are computed based on the overlap of the model outputs with that of the ground-truth.

### C.1.5   ENTITIES: ENUMERATING ENTITIES WITH CONSTRAINTS

We devise 20 questions pertaining to global culture that require consulting a broad range of sources and cannot be accurately answered using a small set of sources on the Web. These questions focus on search intensity. They ask for an exhaustive list of entities within a given category that satisfy all specified criteria. All questions in this category have a fixed reasoning depth, ranging from 2 to 4.

They are centered around real-world events such as award shows, the Olympics, book publications, and similar topics.

**Dataset creation.** We select world events that happen at a regular cadence and design questions that cannot be fully answered using any single webpage or a small set of webpages. Each question in this category requires examining a wide range of items, typically between 80 and 140, and filtering them based on the provided criteria. The final answers consist of 8 to 30 items that meet all specified requirements.

**Ground-Truth Claims.** Each question requires a list of entity names (*claims*) as output. We construct the ground truth using scraping scripts tailored to each world event, followed by manual verification of agent-generated answers, particularly when they are not present in the initial ground truth. After reviewing all agent responses and adding any valid missing answers, we evaluate performance using precision, recall, and F1 scores against the finalized ground truth list.

## C.2 LIVEDRBENCH: ENABLING BENCHMARK UPDATION WITH NOVEL QUESTIONS

We created SCIFACTS queries using Curie Cui et al., by extracting key information from scientific papers and posing queries to use this key information to retrieve the corresponding source paper. Future queries can be similarly generated by applying the same pipeline to newly published papers.

The NOVELDS Peer Retrieval questions can be extended by identifying literature survey tables in newly published papers, and extracting the listed datasets along with their distinguishing features.

For ENTITIES, we include four categories of queries based on distinct world events — movies, book publications, international math Olympiads, and the Olympics. More queries can be generated by selecting a similar event-based category, defining a set criteria, and iteratively scraping all items within that category satisfying the defined criteria.

## C.3 EVALUATION

We structure each task so that it expects a *list of dictionaries* as the output. Each top-level dictionary corresponds to a claim and its keys correspond to sub-claims. For instance, for the material identification task in SCIFACTS, identifying the material name is the claim while identifying the source paper correctly is the sub-claim that supports it. For each key in a dictionary, we define the *agreement* score $s \in [0, 1]$ as a measure of the accuracy of the claim represented by the key's value.

For convenience in evaluation, we include an instruction to output a JSON object in every DR prompt.[1] This avoids the need to parse the output report. However, some DR systems such as Gemini were unable to follow the instruction and still output a report. Therefore, our evaluation pipeline includes an additional step of parsing the keys from a given report. While this can be done by prompting an LLM such as GPT-4, for the purposes of benchmark evaluation, we decided to do it manually to avoid any possibility of error.

**Metrics.** For each benchmark category, we evaluate the correctness (*precision*) of the stated claims and completeness (*recall*) of those claims. We first compute the agreement score $s$ of a generated claim with its ground-truth counterpart (discussed in Section 3.2) using GPT-4o (prompts included in Appendix E). Then, we compute precision and recall metrics as defined in Equation (1). We also report the F1 score combining these two metrics.

However, there is a limitation of the above strategy. While we considered the uniqueness of the answer while creating each DR query, it is possible that multiple correct answers exist. As a result, given only the ground-truth claims, the recall metric may miss correct claims that were not a part of the task's evaluation rubric. Similarly, the precision metric may falsely consider a claim incorrect since it was not present in the task's ground-truth answer. Therefore, we follow a hybrid strategy where we manually evaluate all claims generated by all DR systems under evaluation. If any of the generated claims are correct, we add them to the ground-truth set of claims. After this expansion of the ground-truth, we do an automated evaluation against the expanded ground-truth. Our final benchmark data includes this expanded ground-truth for each question.

---

[1]This format instruction can be easily omitted to allow evaluation without the JSON format requirement.

## D EXAMPLE TASKS FROM LIVEDRBENCH

In this section, we provide examples from each category (Table 2) in the LIVEDRBENCH.

---

**SCIFACTS Materials Prompt**

Find the material(s) that satisfy every one of the listed measured properties.

Instructions
1. In your response, provide a list of JSON object(s) in the following format:

```
[
    {
      "material": "<material name>",
      "inference_basis": "<brief explanation of how the listed
          properties match this material>",
      "paper_title": "<title of the paper>",
      "property_source_table": "<source table from which material
          name, property and property descriptor were identified>",
      "property_source_passage": "<source passage from which
          material name, property and property descriptor were
          identified>"
    }
  ]
```

2. Your inference_basis should reference the key properties (e.g., band gap, absorption coefficient, crystal structure) that led you to that material.
3. You must provide a valid json file even if you don't have the answer.

Measured properties:
- Band gap Eg (room temperature): 3.35 eV
- binding energy: 60 meV
- Lattice constant a (X-ray diffraction XRD): 0.325 nm
- Lattice constant c (X-ray diffraction XRD): 0.521 nm
- Lattice fringe (High-resolution transmission electron microscopy HRTEM): 0.52 nm

---

**SCIFACTS Geo Prompt**

Find the paper(s) that use all of the datasets mentioned below:
Google Earth Engine, Shuttle Radar Topography Mission data, Ground Truth Data, Moderate Resolution Imaging Spectroradiometer (MODIS) Data
Instructions:
1. You must respond with a list JSON object(s) in this format:

```
{
    "paper_title": "<full paper title>"
}
```

2. You must provide a valid json file with empty fields even if you don't have the answer.

NOVELDS Identification Prompt

I'm looking for diffusion based approaches that enable precise control over illumination in images, particularly those that move beyond simple prompt engineering. I'm especially interested in a method that uses a diverse image collection, and incorporates physically grounded constraints—such as consistency under blended lighting conditions—during training, to allow models to respect the structural coherence of light transport. I'm especially curious about methods where the consistency is strong enough to infer structural cues like surface normals purely from illumination variation across synthetic views, without being specifically trained for this property. Find a paper, ideally a published one, that does this.

Once you've found a paper that meets my needs, please provide the information in the following format:

```
{
    'title': <title>,
    'venue': <venue>,
    'year': <year>,
}
```

NOVELDS Identification and Extraction

I'm looking for a GPT-4 generated corpus of decisions rooted in quotidian life - think commuting, family squabbles or career decisions - each tagged against broad socio-psychological dimensions. I'm specifically interested in scenarios where the resolution hinges not on objective correctness, but instead on personal values. How does this corpus reveal GPT-4's implicit generation bias across the various value dimensions in each of the socio-psychological frameworks explored in the corpus?

Provide the an overview of GPT-4's generation bias in the following json format:

```
[
    {
        "framework": <framework_name>,
        "most_biased_dimension": <most_biased_dimension_name>
    },
    ...
]
```

---

NOVELDS Peer Retrieval

I want to compare existing 3D urban segmentation datasets based on real-world scenes that go beyond static 2D maps or flat segmentation overlays, and instead represent urban environments as richly annotated point clouds. How do they compare in terms of data acquisition methods—such as mobile laser scanning, terrestrial laser scanning, aerial laser scanning, and photogrammetry?

For the datasets you find, provide the following information in json format:

```
[
    {
        "name": <name of the resource>,
        "data_aquisition_method": <data acquisition method>,
        "area": <area of the dataset>,
        "scenes": <number of scenes>,
        "points_million": <number of points>,
    },
    ...
]
```

---

ENTITIES Prompt

List all movies released between **2005 and 2011** (inclusive) that have **won an Oscar in any category**, and are **adaptations of works authored (or co-authored) by women**. Please provide the answer in **JSON format**, where each entry includes the following three fields:

- "Movie Name" – the title of the film
- "Original Work" – the name of the source material the movie is based on
- "Author Name" – the name(s) of the female author(s) of the original work

---

FLIGHTS Prompt

In which high-profile flight investigation was military radar data later considered unreliable? Identify the types of radar data that was recorded, what anomalies were found, and why those anomalies raised doubts. Focus on reconstructing a timeline of anomalous readings and critically examine the underlying causes.
For each data type and anomaly, follow this format:

```
[
    {
        "timestamp_utc": "<timestamp>",
        "parameter": "<parameter_name>",
        "recorded_value": "<value>",
        "expected_range_or_behavior": "<expected_range_or_behavior
            >",
        "anomaly_description": "<description_of_anomaly>"
    },
    ...
]
```

> **PRIORART Prompt**
>
> I have the following ideas for a research paper. Can you help identify if this has already been done or implied in full or in parts in other papers?
> Give your answer as a **JSON with three fields: Paper title, link, and connection a field** that quotes exact sentences from the paper and parts of my ideas below to make the case.
>
> Here is the format:
>
> ```
> [
>     {
>         "title": <paper title>,
>         "link": <link to the paper>,
>         "connection": <connection field>
>     }
> ]
> ```
>
> The objective of our paper is to understand optimized prompts (e.g. using black-box or white-box prompt optimization techniques) better in terms of how they differ from human-written forms, and how they help LMs do better predictions, sometimes therefore also enabling jailbreaking, etc. We formulate adversarial attack as a form of controllable text generation. Our method injects attack information during the decoding steps, enabling successful attacks.
>
> The key idea is to automatically generate a "suffix" (a sequence of tokens) to concatenate to the user input (user prompt, task description, etc.). The tokens will be generated one by one left-to-right by maximising a mixed search objective, that uses gradient in the vocabulary and also account for readability of the prompt. Readability in the objective is important especially because a common complaint against optimized prompts is that they often contains illegible text, punctuations, etc.
> Once we have such an optimized readable prompt, we argue that it consists of influential, specific, or rare tokens that help better elicit the user behavior, or in some cases even malicious behavior in case of a jailbreaking attempt. Further, we show that the optimized prompts have distinct internal embeddings (using sparse probing techniques), not just lexical behaviors. Notably, we show results on two new attack scenarios (inducing system prompt leakage and addressing over-censored tasks in LLMs) that have not been explored before.

# E  EVALUATION PROMPTS

Here, we provide the exact prompts used for evaluating the generated outputs. Across all our experiments, we leverage GPT-4o Hurst et al. (2024) as the judge model and compare the generated answer with the ground truth. To ensure the ground truth is comprehensive, we conduct additional human checks and augment it with valid answers that were initially not present. Our benchmark is built on this enriched ground truth set.

For SCIFACTS Materials task we evaluate each key (paper title and material name) present in the generated list of dictionaries against the prompt given below and report the combined metric according to Eq. 1. We use the same evaluation prompt to evaluate the paper title in SCIFACTS Geo task.

> **Prompt for evaluating each key in SciFacts**
>
> You are an evaluator. Given a Golden Answer and a Predicted Answer, check whether the Predicted Answer is correct. The prediction is correct if it fully aligns with and contains all the key information present in the Golden Answer. Respond with True if the prediction is correct and False otherwise.
> Golden Answer: {golden-answer}
> Predicted Answer: {predicted-answer}

For the NOVELDS and FLIGHTS queries in which both the prediction and ground truth consist of lists of dictionaries, we begin by identifying the corresponding ground-truth dictionary for each predicted dictionary. This is done using a set of primary keys to be used for matching, using the following prompt:

---

**NOVELDS and FLIGHTS prompt to match a predicted dictionary against a list of ground truth dictionaries using a set of primary keys**

You are given a list of dictionaries (the ground truth) and a single predicted dictionary (the model output). Your task is to determine which dictionary in the ground truth list corresponds with the predicted dictionary, using a set of primary keys to identify the best match.

- You will be a given a list of primary keys that can be used to compare the dictionaries. If a primary key is not present in two dictionaries, make sure to use the other primary keys to determine the match.
- Do not consider ANY other fields in the dictionaries except for the primary keys. - If the value is a string, consider them equivalent if they are similar in meaning, or if one is a subset of the other.
- Match based on semantic similarity or if one is a subset of the other, not exact string equality.
- Acronyms, shortened names, or partial names are equivalent to their expanded forms.
- "BERT" and "BERT: Pre-training of Deep Bidirectional Transformers for Language Understanding" → equivalent
- "ICLR" and "International Conference on Learning Representations" → equivalent
- A prediction may include extra info (e.g., a year or qualifier). If the ground truth is a subset and meaning is preserved, treat them as equivalent. If the ground truth includes a qualifier, the prediction's qualifier must match if present.
- You may also be provided with a set of extra evaluation notes that provide more specific evaluation criteria. These extra rules may override the general rules above.
- If after all this no match exists, return an empty dictionary.

Now compare the following:

Ground Truth List:
{list-gt}

Model Output:
{dict-pred}

Primary Keys:
{primary-keys}

Extra Evaluation Notes:
{note}

Your output should be the dictionary from the ground truth list that best matches the model output, or an empty dictionary if no match is found.

IMPORTANT:
- Do NOT explain your reasoning.
- Do NOT include any extra text.
- Only output the final dictionary as a JSON object. Do not wrap it in markdown or say anything else.

---

With a corresponding ground-truth dictionary obtained for each predicted dictionary in the NOVELDS and FLIGHTS queries, we evaluate the entire predicted dictionary against the ground truth dictionary, rather than key by key. This is because related sub-claims provide important context, and evaluating keys individually can lead to under-scoring by the LLM judge. The exact prompt used for evaluation is provided below:

NOVELDS and FLIGHTS prompt for evaluating a predicted dictionary against a ground truth dictionary

You are evaluating whether two dictionaries refer to the same information. Your task is to determine whether each corresponding value is equivalent.

For each key-value pair in the ground truth dictionary:
- If the key is absent in the model output dictionary, it is not equivalent.
- If the value is a number (integer or float), they are equivalent within a 1% margin of error.
- If the value is a string, consider them equivalent if they are similar in meaning, or if one is a subset of the other.
- Acronyms or shortened names are equivalent to their expanded forms.
- "BERT" and "BERT: Pre-training of Deep Bidirectional Transformers for Language Understanding" → equivalent
- "ICLR" and "International Conference on Learning Representations" → equivalent
- "CNN" and "ResNet" → equivalent (both are convolutional neural networks)
- "Adam" and "SGD" → not equivalent (different optimizers)
- "0.8" and "0.80" → equivalent numerically
- If the value is a URL, they must be exactly the same, or a shortened version of the same URL.
- If the value is a list, check if every element in the ground truth list is present in the model output list, regardless of order. Follow the same rules for strings as above.
- If the value is a dictionary, check if every key-value pair in the ground truth dictionary is present in the model output dictionary, regardless of order. Again, follow the same rules for strings as above.
- Some values may depend on information from other keys in the dictionary. Use such context to determine whether two values are equivalent, even if they are not identical on their own.
- Ignore trivial formatting differences such as casing, whitespace, or common abbreviations if the meaning is preserved.

Now compare the following dictionaries:

Ground Truth:
{dict-gt}

Model Output:
{dict-pred}

Your output should be a JSON object with the same keys as the input dictionaries. For each key, the value should be:
- 3: The predicted value closely matches the ground-truth, preserving almost all important information with minimal or no loss of meaning.
- 2: The predicted value captures the main idea of the ground-truth but omits key details or has minor inaccuracies.
- 1: The predicted value shows some overlap with the ground-truth but misses most of the essential meaning.
- 0: The predicted value is largely incorrect, unrelated, or does not correspond meaningfully to the ground-truth value.

IMPORTANT:
- Do NOT explain your reasoning.
- Do NOT include any extra text.
- Only output the final JSON object. Do not wrap it in markdown or say anything else.

Finally, we evaluate PRIORART and ENTITIES using the prompt below. Here, the LLM judge is required to check if the predicted string is present in the list of ground truths (paper titles and entities like names of people or movies). We ensure the minor differences in formatting are ignored.

PRIORART and ENTITIES prompt to match a predicted string against a list of ground truth strings

You are evaluating whether a predicted string is present in a ground-truth list of strings.

- Allow for minor differences in formatting, such as casing, whitespace, or common abbreviations.
- If found, return the matching string from the ground truth list in JSON format:
{ "match": "matched-string" }
- If not, return an empty dictionary.

Now compare the following:

Ground Truth List:
{gt-list}

Predicted String:
{pred-str}

IMPORTANT:
- Do NOT explain your reasoning.
- Do NOT include any extra text.
- Only output the JSON object. Do not wrap it in markdown or say anything else.

## F  JSON EXTRACTION PROMPT

Prompt for extracting JSON fields from a DR report

You are a JSON extraction assistant.

You are given:
1. A JSON structure which can be either a list of values, or a list of dictionaries with keys.
2. A textual report that contains information corresponding to the JSON structure.

Your task:
- Extract the value(s) from the report for each element or key in the JSON.
- Fill in the JSON with the extracted values.
- If a value cannot be found in the report, set it as null.
- Respond **only** with the completed JSON, no extra text or explanation.

Process the following:

Report:

{report}
JSON structure:

{json_schema}

# G TRACE ANALYSIS PROMPTS

> **Prompt to get nescessary queries given a DR question**
>
> Given a question, extract the important search queries necessary to answer it using web search. Focus on queries without which the question cannot be answered fully.
> Output as JSON with two categories:
> - characteristic-searches: Search queries related to the unique aspects of the question. These will be used to find a source paper/report that contains the answers to the question.
> - extractive-searches: You will see that the question asks for specific fields to be extracted into a JSON object. Come up with search queries aimed at extracting specific this information from the source paper/report. Think of searches for each JSON field and sub-field that the question asks for, and nothing else.
>
> **Example**
> **Question:** Find a paper which introduces an image dataset designed to promote the development of autonomous deep learning systems for detecting and classifying waste in complex, real-world, in-situ environments.
> What are the top 10 classes/categories by number of annotations in the dataset? Provide them in the format:
>
> ```
> [
>     { "class-category": <class-category-name>,},
>     { "class-category": <class-category-name>,},
>     ...
> ]
> ```
>
> What baselines were used by the dataset? Provide the baselines used by the dataset in the format:
>
> ```
> [
>     {
>         'dataset': <dataset-name>,
>         'model': <model-name>,
>         <test-1>: <test-1-value>,
>         ...
>     },
> ]
> ```
>
> **Output:**
>
> ```
> {
>     "characteristic-searches": [
>         "image dataset for waste classification",
>         "autonomous deep learning system for waste detection",
>         "real-world, in-situ waste data collection",
>     ]
> },
> {
>     "extractive-searches": [
>         "top classes by number of annotations",
>         "baseline models and benchmark results",
>         "evaluation metrics used"
>     ]
> }
> ```
>
> **Question**
> {question}
> **IMPORTANT**
> - Ensure the search terms are as comprehensive and explicit as possible. - Avoid vague or generic terms that don't add concrete search value. - Avoid redundancy; each search term should cover a distinct aspect.

Prompt for checking if a query is present in a DR model's trace

Given a search and reasoning trace from a language model, figure out if a particular query was invoked by the model during its reasoning process. It doesn't matter if the model got the answer wrong, just figure out if it was invoked or not. It also doesn't need to be an exact match, but every important aspect of the query should be present in the invoked query.
**Output Format:**

```
{
    "excerpt": <the verbatim excerpt from the trace that indicates
               the query was invoked, or "" if not found>,
    "invoked": true/false,
}
```
**Trace:**
{trace}

**Query:**
{query}

**IMPORTANT:**
- Only output the final dictionary as a JSON object.
- Do NOT explain your reasoning.

---

**Prompt to get dependent and independent queries from a DR model's trace**

You are a subquery analysis assistant.

You are given:
- A **trace**: the reasoning steps or prior context that has already been carried out.
- A **paragraph**: a passage containing one or more possible queries.

Your task is:
1. **Enumerate all subqueries** in the paragraph. A subquery is any distinct request for information or lookup implied by the text. There may be multiple subqueries, or none at all.
2. For each subquery, decide whether it is **Independent** or **Dependent**:
- **Dependent**: The subquery builds on, refers to, or modifies something in the trace (e.g., a comparison, continuation, or follow-up based on what the trace already covered).
- **Independent**: The subquery initiates a new line of inquiry that does not rely on the trace.

Step 1: Enumerate Subqueries
Extract all distinct subqueries from the paragraph. Each subquery should be a standalone question or request for information.

Step 2: Analyse with respect to the trace
For each subquery, analyse its dependency on the trace. If a trace is empty, then all subqueries are "Independent".

Step 3: Classify Subqueries
Finally, for each subquery, assign a dependency label:
- "Dependent"
- "Independent"

Output Format

```
[
    {
        "subquery": "[text of the subquery]",
        "dependency_analysis": "[analysis]",
        "dependency": "Independent/Dependent",
    },
    {
        "subquery": "[text of the subquery]",
        "dependency_analysis": "[analysis]",
        "dependency": "Independent/Dependent",
    }
]
```
Trace:
{trace}

Paragraph:
{paragraph}

---

**Prompt to count the number of branches in a given DR trace**

You are an expert in analyzing research investigation traces. Your task is to read the provided trace and identify how many distinct research *branches* or lines of inquiry the author explores. A new branch is a distinct hypothesis, search direction, or subgoal. Count how many times the researcher changes focus, searches for different types of information, or explores parallel possibilities.

Output ONLY the number of branches as an integer.

---

**Prompt to count the number of backtracking events in a given DR trace**

You are an expert in analyzing research investigation traces. Your task is to read the provided trace and count how many times the researcher *backtracks* — that is, revisits or reconsiders previous ideas, changes direction after realizing an approach is suboptimal, or discards a line of inquiry.

Look for transitions like:
- 'I first thought... but then...'
- 'Initially considered... later found...'
- 'Switched from... to...'

There may or may not be any transition phrases, just count the number of ideas or directions changed.

Count only actual *changes in direction or reconsiderations*. Output ONLY the number of backtracking steps as an integer.

---

## H  QUALITATIVE ANALYSIS

We present a few qualitative examples showing the necessary queries that the DR models failed to invoke.

---

**Example Question 1**

I want a paper that introduces a dataset and method to test image editing models. In my setting, I don't have explicit instructions to edit the images, but I have similar edits done on another image and I want the edit to carry over. It would be great if the method is efficient, and does not require any optimization whatsoever.

Once you've found a relevant paper, please provide the following information in JSON format:

```
{
    "paper_title": "<title_of_the_paper>",
    "is_published": <true/false>,
    "venue": "<venue_name>",
    "year": <publication_year>,
    "dataset_link": "<link_to_the_dataset>"
}
```

---

For this question, only Gemini DR invoked a query for *"method for transferring edits between images"*, while OpenAI DR, Perplexity DR, and both variants of DeepResearcher failed to invoke it.

---

**Example Question 2**

Identify a publicly available dataset of long-form in-the-wild videos that are segmented into scenes and richly annotated with both content descriptions and user engagement signals. The dataset should include multimodal information and support analysis at the video level as well as the scene level.

Once you've found a dataset paper, provide its information in the following JSON format:

```
{
    "title": <title of the dataset paper>,
    "is_published": <true/false>,
    "venue": <venue>, // eg: CVPR
    "year": <year of publication>,
    "dataset_url": <link to the dataset>, // if available, else null
}
```

---

For this question, only OpenAI DR and Perplexity DR invoked a query for *"in-the-wild video dataset with scene segmentation"*, while Gemini DR, and both variants of DeepResearcher failed to invoke it.

## I    ADDITIONAL RESULTS

| Subset | OpenAI Deep Research | | | Perplexity Pro Research | | | Gemini Deep Research (2.5 Pro) | | |
|---|---|---|---|---|---|---|---|---|---|
| | Prec | Rec | F1 | Prec | Rec | F1 | Prec | Rec | F1 |
| SCIFACTS Geo | 0.715 | 0.728 | 0.721 | 0.263 | 0.144 | 0.186 | 0.405 | 0.259 | 0.316 |
| SCIFACTS Materials | 0.313 | 0.316 | 0.314 | 0.264 | 0.105 | 0.150 | 0.072 | 0.013 | 0.022 |
| NOVELDS Identification | 0.667 | 0.667 | 0.667 | 0.625 | 0.625 | 0.625 | 0.375 | 0.375 | 0.375 |
| NOVELDS Identification and Extraction | 0.559 | 0.428 | 0.462 | 0.411 | 0.351 | 0.358 | 0.390 | 0.284 | 0.319 |
| NOVELDS Peer Retrieval | 0.849 | 0.498 | 0.609 | 0.535 | 0.248 | 0.322 | 0.558 | 0.241 | 0.334 |
| PRIORART | 0.714 | 0.473 | 0.552 | 0.701 | 0.349 | 0.430 | 0.174 | 0.143 | 0.133 |
| ENTITIES | 0.813 | 0.534 | 0.603 | 0.673 | 0.373 | 0.447 | 0.494 | 0.3 | 0.338 |
| FLIGHTS | 0.525 | 0.530 | 0.523 | 0.385 | 0.358 | 0.369 | 0.243 | 0.264 | 0.252 |
| **Average** | 0.644 | 0.522 | 0.556 | 0.482 | 0.319 | 0.361 | 0.339 | 0.235 | 0.261 |

Table 6: Precision, recall, and F1 scores on LIVEDRBENCH for Deep Research models. Claim agreement score is evaluated by the authors.

### I.1    CATEGORY-WISE RESULTS

We present category-wise results for the DR models and baselines in Tables 7–10. Among the baseline non-DR models, reasoning models struggle when their reasoning is limit to ≈30 seconds, underscoring the complexity of LiveDRBench queries – requiring intensive compute and planning capabilities. In particular, all reasoning models obtain a F1 score of zero for the SCIFACTS Materials dataset. The non-reasoning models perform even worse, due to their limited ability to plan, conduct iterative searches, and backtrack. Interestingly, OpenAI's o4-mini performs comparably or even better than Perplexity DR and Gemini DR in many of the task categories, yet significantly behind OpenAI DR.

### I.2    HUMAN EVALUATION

To check for errors due to GPT-4o based agreement scores, we also present human-evaluated precision, recall, and F1 scores across all categories in Table 6. Overall, we find that the metrics computed using

| Model | Precision | Recall | F1 |
|---|---|---|---|
| OpenAI DR | 0.715 | 0.728 | **0.721** |
| Perplexity DR | 0.263 | 0.144 | 0.186 |
| Gemini DR | 0.405 | 0.259 | 0.316 |
| DeepResearcher + DS Qwen 32B | 0.000 | 0.000 | 0.000 |
| DeepResearcher + GPT-4.1 | 0.000 | 0.000 | 0.000 |
| OpenAI o4-mini | 0.138 | 0.104 | 0.114 |
| Sonar Reasoning | 0.000 | 0.000 | 0.000 |
| Gemini 2.5 Pro | 0.273 | 0.200 | **0.201** |
| OpenAI GPT-4.1 | 0.017 | 0.021 | 0.019 |
| Sonar Pro | 0.000 | 0.000 | 0.000 |
| Gemini 2.5 Flash | 0.059 | 0.059 | **0.059** |

Table 7: Comparison of DR models on the SCI-FACTS Geo tasks.

| Model | Precision | Recall | F1 |
|---|---|---|---|
| OpenAI DR | 0.313 | 0.316 | **0.314** |
| Perplexity DR | 0.264 | 0.105 | 0.150 |
| Gemini DR | 0.072 | 0.013 | 0.022 |
| DeepResearcher + DS Qwen 32B | 0.000 | 0.000 | 0.000 |
| DeepResearcher + GPT-4.1 | 0.000 | 0.000 | 0.000 |
| OpenAI o4-mini | 0.000 | 0.000 | 0.000 |
| Sonar Reasoning | 0.000 | 0.000 | 0.000 |
| Gemini 2.5 Pro | 0.000 | 0.000 | 0.000 |
| OpenAI GPT-4.1 | 0.000 | 0.000 | 0.000 |
| Sonar Pro | 0.000 | 0.000 | 0.000 |
| Gemini 2.5 Flash | 0.000 | 0.000 | 0.000 |

Table 8: Comparison of DR models on the SCI-FACTS Materials tasks.

the claim agreement scores determined through manual verification by the authors closely match that of the results in Table 3. This indicates that our benchmark can be used to reliably evaluate any new model using the evaluation prompts in Appendix E.

| Model | Precision | Recall | F1 |
|---|---|---|---|
| OpenAI DR | 0.542 | 0.546 | **0.540** |
| Perplexity DR | 0.389 | 0.347 | 0.362 |
| Gemini DR | 0.249 | 0.276 | 0.261 |
| DeepResearcher + DS Qwen 32B | 0.120 | 0.073 | 0.090 |
| DeepResearcher + GPT-4.1 | 0.044 | 0.022 | 0.027 |
| OpenAI o4-mini | 0.340 | 0.282 | **0.304** |
| Sonar Reasoning | 0.237 | 0.168 | 0.183 |
| Gemini 2.5 Pro | 0.228 | 0.210 | 0.215 |
| OpenAI GPT-4.1 | 0.200 | 0.138 | 0.160 |
| Sonar Pro | 0.216 | 0.187 | **0.194** |
| Gemini 2.5 Flash | 0.182 | 0.105 | 0.117 |

Table 9: Comparison of DR models and baselines on the FLIGHTS tasks.

| Model | Precision | Recall | F1 |
|---|---|---|---|
| OpenAI DR | 0.813 | 0.534 | **0.603** |
| Perplexity DR | 0.673 | 0.373 | 0.447 |
| Gemini DR | 0.494 | 0.3 | 0.338 |
| DeepResearcher + DS Qwen 32B | 0.307 | 0.046 | 0.076 |
| DeepResearcher + GPT-4.1 | 0.303 | 0.035 | 0.058 |
| OpenAI o4-mini | 0.265 | 0.082 | 0.115 |
| Sonar Reasoning | 0.199 | 0.04 | 0.064 |
| Gemini 2.5 Pro | 0.254 | 0.120 | **0.151** |
| OpenAI GPT-4.1 | 0.217 | 0.0725 | **0.074** |
| Sonar Pro | 0.075 | 0.035 | 0.042 |
| Gemini 2.5 Flash | 0.153 | 0.044 | 0.064 |

Table 10: Comparison of DR models on the EN-TITIES tasks.

## I.3 SENSITIVITY ANALYSIS

To assess evaluation robustness, we repeat the LLM-as-judge evaluation five times with temperature 0.3 and report the standard deviation across runs. Tables 14–19 summarize results using GPT-4o and Qwen3-32B (non-thinking) as judges. Across our experiments, variance remains low (typically $\leq 0.02$ absolute), with overall standard deviation ranging from 0.004 to 0.012. This stability is expected because the judge task consists of verification against ground-truth claims. Replacing GPT-4o with Qwen3-32B yields similarly low variance, indicating that the evaluation framework is robust to judge model choice.

## I.4 IMPACT OF REASONING STRENGTH

We evaluate non-DR models with reasoning set to *high*, with and without improved prompts (Table 20). Our primary objective is to demonstrate that *reasoning intensity* is a key driver of performance on LiveDRBench. To further substantiate this claim, we evaluate non-DR models with reasoning explicitly set to *high*, both with default prompting and with an improved (efficient) prompt designed to encourage structured claim extraction and verification. Results are shown in Table 20. We observe

| Model | Precision | Recall | F1 |
|---|---|---|---|
| OpenAI DR | 0.667 | 0.667 | **0.667** |
| Perplexity DR | 0.633 | 0.633 | 0.633 |
| Gemini DR | 0.400 | 0.400 | 0.400 |
| DeepResearcher + DS Qwen 32B | 0.167 | 0.167 | 0.167 |
| DeepResearcher + GPT-4.1 | 0.233 | 0.233 | 0.233 |
| OpenAI o4-mini | 0.467 | 0.467 | **0.467** |
| Sonar Reasoning | 0.200 | 0.200 | 0.200 |
| Gemini 2.5 Pro | 0.444 | 0.444 | 0.444 |
| OpenAI GPT-4.1 | 0.333 | 0.333 | 0.333 |
| Sonar Pro | 0.233 | 0.233 | 0.233 |
| Gemini 2.5 Flash | 0.378 | 0.378 | **0.378** |

Table 11: Comparison of DR models and baselines on the NOVELDS Identification tasks.

| Model | Precision | Recall | F1 |
|---|---|---|---|
| OpenAI DR | 0.795 | 0.480 | **0.585** |
| Perplexity DR | 0.494 | 0.248 | 0.311 |
| Gemini DR | 0.551 | 0.249 | 0.338 |
| DeepResearcher + DS Qwen 32B | 0.083 | 0.028 | 0.042 |
| DeepResearcher + GPT-4.1 | 0.067 | 0.019 | 0.029 |
| OpenAI o4-mini | 0.652 | 0.254 | **0.345** |
| Sonar Reasoning | 0.083 | 0.009 | 0.017 |
| Gemini 2.5 Pro | 0.334 | 0.140 | 0.196 |
| OpenAI GPT-4.1 | 0.465 | 0.196 | **0.276** |
| Sonar Pro | 0.231 | 0.083 | 0.122 |
| Gemini 2.5 Flash | 0.229 | 0.163 | 0.190 |

Table 12: Comparison of DR models and baselines on the NOVELDS Peer Retrieval tasks.

| Model | Precision | Recall | F1 |
|---|---|---|---|
| OpenAI DR | 0.694 | 0.463 | **0.539** |
| Perplexity DR | 0.689 | 0.339 | 0.419 |
| Gemini DR | 0.106 | 0.096 | 0.082 |
| DeepResearcher + DS Qwen 32B | 0.525 | 0.136 | 0.199 |
| DeepResearcher + GPT-4.1 | 0.412 | 0.135 | 0.180 |
| OpenAI o4-mini | 0.529 | 0.179 | 0.254 |
| Sonar Reasoning | 0.230 | 0.149 | 0.166 |
| Gemini 2.5 Pro | 0.309 | 0.322 | **0.307** |
| OpenAI GPT-4.1 | 0.362 | 0.297 | **0.311** |
| Sonar Pro | 0.180 | 0.100 | 0.113 |
| Gemini 2.5 Flash | 0.272 | 0.228 | 0.227 |

Table 13: Comparison of DR models and baselines on the PRIORART tasks.

substantial improvements when stronger reasoning and improved prompting are applied. For example, o3 improves from 0.539 to 0.666 F1 with an efficient prompt. These results support our hypothesis that higher reasoning intensity significantly improves performance on claims-based deep research evaluation.

### I.5 VARIANCE AND STATISTICAL TESTS

To address concerns regarding robustness and statistical validity, we conduct additional analyses covering (i) alignment with human evaluation, (ii) per-example variability (iii) confidence intervals, and (iv) variance across multiple DR runs.

**(i) Alignment with Human Evaluation.** We evaluate agreement between GPT-4o and human evaluation using (1) Cohen's $\kappa$ (following DeepResearchGym (Coelho et al., 2025)) and (2) Mean Absolute Error (MAE). Human scores (Table 6) are compared with GPT-4o scores (Table 3). Three authors independently performed the human evaluation. Cohen's $\kappa$ ranges from 0 (chance agreement) to 1 (perfect agreement). Table 21 shows strong agreement (high $\kappa$) and low deviation (low MAE) between GPT-4o and human evaluators establishing the efficacy of our model based evaluation. We also compute the Mean Absolute Error (MAE) between Human and GPT-4o Judge and present the results in Table 22 – demonstrating low average MAE across the board.

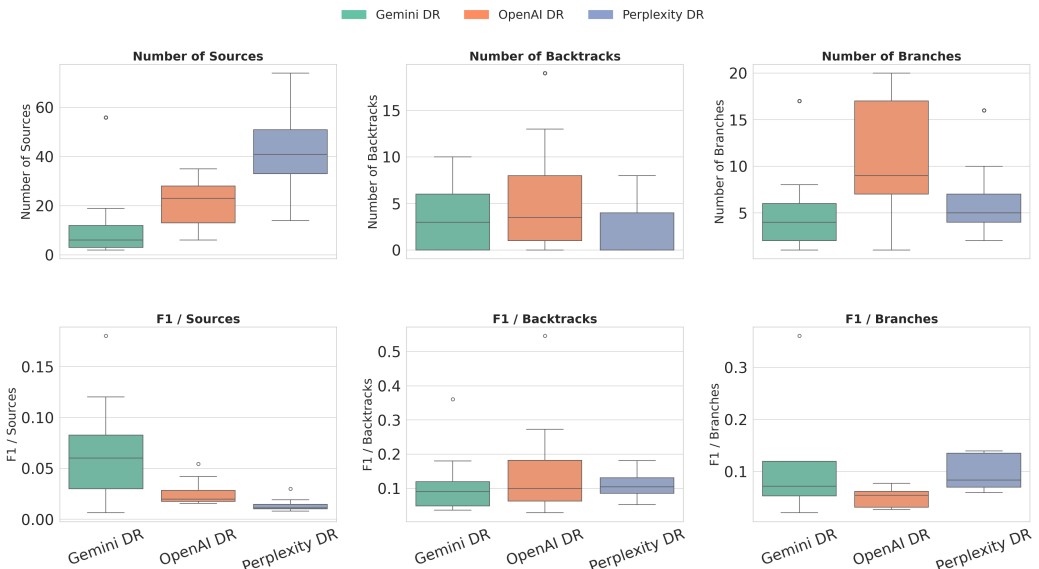

Figure 5: (Top) Box Plots illustrating the number of **sources** referred, **branches** considered, and **backtracking** events occurred against F1 score on the NOVELDS subset using three DR models: Gemini DR, OpenAI DR (OAI DR), Perplexity DR. (Bottom) F1 Efficiency per event.

Table 14: Std. dev. across 5 GPT-4o evaluation runs for OpenAI DR (temp=0.3).

| Subset | Precision | Recall | F1 |
|---|---|---|---|
| SciFacts Materials | $0.312 \pm 0.000$ | $0.316 \pm 0.000$ | $0.314 \pm 0.000$ |
| SciFacts Geo | $0.715 \pm 0.000$ | $0.707 \pm 0.026$ | $0.711 \pm 0.026$ |
| NovelDS Identification | $0.667 \pm 0.000$ | $0.667 \pm 0.000$ | $0.667 \pm 0.000$ |
| NovelDS Identi-Extraction | $0.546 \pm 0.022$ | $0.466 \pm 0.019$ | $0.487 \pm 0.020$ |
| NovelDS Peer | $0.791 \pm 0.012$ | $0.477 \pm 0.008$ | $0.581 \pm 0.009$ |
| PriorArt | $0.694 \pm 0.000$ | $0.463 \pm 0.000$ | $0.539 \pm 0.000$ |
| Entities | $0.791 \pm 0.012$ | $0.517 \pm 0.008$ | $0.584 \pm 0.010$ |
| Flights | $0.560 \pm 0.029$ | $0.556 \pm 0.022$ | $0.553 \pm 0.025$ |
| Overall Avg | $0.634 \pm 0.009$ | $0.521 \pm 0.010$ | $0.554 \pm 0.011$ |

**(ii) Per-Example Variability.** We compute per-example standard errors across all DR models evaluated on LiveDRBench (Table 23). Standard errors remain low across models and metrics, indicating stable aggregate performance despite task difficulty.

**(iii) Confidence Intervals.** We further report 95% confidence intervals under (a) Gaussian assumptions and (b) bootstrap resampling (Tables 25). Both approaches yield consistent intervals, confirming that our inferences are not sensitive to distributional assumptions.

**(iv) Variance Across DR Runs.** After API availability, we performed multiple runs of O3-Deep-Research on a LiveDRBench subset (Entities). We follow recommended prompts, default reasoning settings, and set `max_tool_calls=500`. Results are shown in Table 26.

### I.6 METRIC DESIGN ABLATION

We consider the strict version of our precision and recall metric for evaluation in this section. Specifically, in this setting, we compute a min over all sub-claims predicted subclaims. In SciFacts Geo, PriorArt, and Entities, there are only main claims with no subclaims. In SciFacts Materials, there is a single subclaim. In either case, computing a min over the subclaims has no effect on the evaluation. We present these alternative evaluation results in Tables 27, 29, 28. The trend among DR models remains consistent under the strict min metric as well.

Table 15: Std. dev. across 5 GPT-4o evaluation runs for Perplexity DR (temp=0.3).

| Subset | Precision | Recall | F1 |
|---|---|---|---|
| SciFacts Materials | $0.211 \pm 0.012$ | $0.105 \pm 0.000$ | $0.141 \pm 0.000$ |
| SciFacts Geo | $0.263 \pm 0.000$ | $0.145 \pm 0.000$ | $0.187 \pm 0.000$ |
| NovelDS Identification | $0.633 \pm 0.000$ | $0.633 \pm 0.000$ | $0.633 \pm 0.000$ |
| NovelDS Identi-Extraction | $0.369 \pm 0.017$ | $0.395 \pm 0.018$ | $0.378 \pm 0.017$ |
| NovelDS Peer | $0.553 \pm 0.033$ | $0.261 \pm 0.008$ | $0.332 \pm 0.013$ |
| PriorArt | $0.689 \pm 0.000$ | $0.339 \pm 0.000$ | $0.419 \pm 0.000$ |
| Entities | $0.612 \pm 0.000$ | $0.343 \pm 0.000$ | $0.419 \pm 0.000$ |
| Flights | $0.415 \pm 0.026$ | $0.374 \pm 0.024$ | $0.388 \pm 0.024$ |
| Overall Avg | $0.468 \pm 0.011$ | $0.324 \pm 0.006$ | $0.362 \pm 0.007$ |

Table 16: Std. dev. across 5 GPT-4o evaluation runs for Gemini DR (temp=0.3).

| Subset | Precision | Recall | F1 |
|---|---|---|---|
| SciFacts Materials | $0.072 \pm 0.000$ | $0.013 \pm 0.000$ | $0.021 \pm 0.000$ |
| SciFacts Geo | $0.405 \pm 0.000$ | $0.259 \pm 0.000$ | $0.316 \pm 0.000$ |
| NovelDS Identification | $0.400 \pm 0.000$ | $0.400 \pm 0.000$ | $0.400 \pm 0.000$ |
| NovelDS Identi-Extraction | $0.430 \pm 0.017$ | $0.339 \pm 0.016$ | $0.362 \pm 0.015$ |
| NovelDS Peer | $0.526 \pm 0.020$ | $0.241 \pm 0.007$ | $0.326 \pm 0.010$ |
| PriorArt | $0.106 \pm 0.000$ | $0.096 \pm 0.000$ | $0.082 \pm 0.000$ |
| Entities | $0.439 \pm 0.000$ | $0.304 \pm 0.000$ | $0.337 \pm 0.000$ |
| Flights | $0.277 \pm 0.011$ | $0.313 \pm 0.015$ | $0.293 \pm 0.013$ |
| Overall Avg | $0.332 \pm 0.006$ | $0.246 \pm 0.005$ | $0.267 \pm 0.005$ |

Table 17: Std. dev. across 5 Qwen3-32B (non-thinking) evaluation runs for OpenAI DR (temp=0.3).

| Subset | Precision | Recall | F1 |
|---|---|---|---|
| SciFacts Materials | $0.332 \pm 0.028$ | $0.355 \pm 0.055$ | $0.343 \pm 0.000$ |
| SciFacts Geo | $0.715 \pm 0.000$ | $0.728 \pm 0.000$ | $0.721 \pm 0.000$ |
| NovelDS Identification | $0.667 \pm 0.000$ | $0.667 \pm 0.000$ | $0.667 \pm 0.000$ |
| NovelDS Identi-Extraction | $0.506 \pm 0.009$ | $0.415 \pm 0.008$ | $0.438 \pm 0.009$ |
| NovelDS Peer | $0.789 \pm 0.008$ | $0.476 \pm 0.003$ | $0.580 \pm 0.005$ |
| PriorArt | $0.714 \pm 0.000$ | $0.473 \pm 0.000$ | $0.552 \pm 0.000$ |
| Entities | $0.811 \pm 0.000$ | $0.532 \pm 0.000$ | $0.600 \pm 0.000$ |
| Flights | $0.589 \pm 0.023$ | $0.572 \pm 0.016$ | $0.574 \pm 0.019$ |
| Overall Avg | $0.640 \pm 0.008$ | $0.527 \pm 0.010$ | $0.559 \pm 0.004$ |

Table 18: Std. dev. across 5 Qwen3-32B (non-thinking) evaluation runs for Perplexity DR (temp=0.3).

| Subset | Precision | Recall | F1 |
|---|---|---|---|
| SciFacts Materials | $0.264 \pm 0.000$ | $0.164 \pm 0.000$ | $0.202 \pm 0.000$ |
| SciFacts Geo | $0.363 \pm 0.011$ | $0.233 \pm 0.036$ | $0.284 \pm 0.036$ |
| NovelDS Identification | $0.633 \pm 0.000$ | $0.633 \pm 0.000$ | $0.633 \pm 0.000$ |
| NovelDS Identi-Extraction | $0.358 \pm 0.016$ | $0.385 \pm 0.018$ | $0.369 \pm 0.016$ |
| NovelDS Peer | $0.565 \pm 0.008$ | $0.264 \pm 0.003$ | $0.337 \pm 0.004$ |
| PriorArt | $0.704 \pm 0.006$ | $0.351 \pm 0.004$ | $0.432 \pm 0.005$ |
| Entities | $0.666 \pm 0.000$ | $0.351 \pm 0.000$ | $0.431 \pm 0.000$ |
| Flights | $0.447 \pm 0.042$ | $0.381 \pm 0.036$ | $0.405 \pm 0.038$ |
| Overall Avg | $0.500 \pm 0.010$ | $0.345 \pm 0.012$ | $0.387 \pm 0.012$ |

Table 19: Std. dev. across 5 Qwen3-32B (non-thinking) evaluation runs for Gemini DR (temp=0.3).

| Subset | Precision | Recall | F1 |
|---|---|---|---|
| SciFacts Materials | $0.075 \pm 0.003$ | $0.013 \pm 0.000$ | $0.022 \pm 0.001$ |
| SciFacts Geo | $0.405 \pm 0.000$ | $0.272 \pm 0.005$ | $0.326 \pm 0.005$ |
| NovelDS Identification | $0.400 \pm 0.000$ | $0.400 \pm 0.000$ | $0.400 \pm 0.000$ |
| NovelDS Identi-Extraction | $0.397 \pm 0.016$ | $0.306 \pm 0.019$ | $0.333 \pm 0.016$ |
| NovelDS Peer | $0.552 \pm 0.006$ | $0.250 \pm 0.002$ | $0.339 \pm 0.003$ |
| PriorArt | $0.181 \pm 0.000$ | $0.147 \pm 0.000$ | $0.141 \pm 0.000$ |
| Entities | $0.451 \pm 0.001$ | $0.307 \pm 0.001$ | $0.342 \pm 0.001$ |
| Flights | $0.280 \pm 0.023$ | $0.314 \pm 0.030$ | $0.295 \pm 0.026$ |
| Overall Avg | $0.343 \pm 0.006$ | $0.251 \pm 0.007$ | $0.275 \pm 0.007$ |

Table 20: Non-DR models with high reasoning.

| Model | Precision | Recall | F1 |
|---|---|---|---|
| o3 (efficient prompt) | 0.800 | 0.616 | 0.666 |
| o3 | 0.645 | 0.492 | 0.539 |
| o4_mini (efficient prompt) | 0.720 | 0.341 | 0.402 |

Table 21: Cohen's $\kappa$ between Human and GPT-4o.

| Comparison | Precision | Recall | F1 |
|---|---|---|---|
| OpenAI vs. Perplexity DR | 0.98 | 0.98 | 0.96 |
| Perplexity vs. Gemini DR | 0.86 | 0.92 | 0.86 |
| Gemini vs. OpenAI DR | 0.98 | 0.96 | 0.96 |

Table 22: MAE between Human and GPT-4o.

| System | Precision | Recall | F1 |
|---|---|---|---|
| OpenAI DR | 0.020 | 0.012 | 0.014 |
| Perplexity DR | 0.022 | 0.022 | 0.021 |
| Gemini DR | 0.035 | 0.027 | 0.028 |

Table 23: Per-example variability on LiveDRBench.

| Model | Precision $\pm$ SE | Recall $\pm$ SE | F1 $\pm$ SE |
|---|---|---|---|
| OpenAI DR | $0.633 \pm 0.031$ | $0.523 \pm 0.034$ | $0.555 \pm 0.032$ |
| Perplexity DR | $0.466 \pm 0.037$ | $0.317 \pm 0.029$ | $0.355 \pm 0.030$ |
| Gemini DR | $0.335 \pm 0.031$ | $0.240 \pm 0.025$ | $0.263 \pm 0.025$ |
| DeepResearcher + DS Qwen 32B | $0.166 \pm 0.024$ | $0.069 \pm 0.013$ | $0.086 \pm 0.015$ |
| DeepResearcher + GPT4.1 | $0.163 \pm 0.025$ | $0.076 \pm 0.014$ | $0.088 \pm 0.015$ |

Table 24: 95% Confidence Intervals assuming Gaussian distribution.

| Model | Precision CI | Recall CI | F1 Score CI |
|---|---|---|---|
| OpenAI DR | $0.571 - 0.695$ | $0.456 - 0.588$ | $0.484 - 0.610$ |
| Perplexity DR | $0.393 - 0.538$ | $0.253 - 0.368$ | $0.283 - 0.401$ |
| Gemini DR | $0.277 - 0.399$ | $0.191 - 0.288$ | $0.208 - 0.306$ |
| DeepResearcher + DS Qwen 32B | $0.118 - 0.214$ | $0.044 - 0.095$ | $0.057 - 0.115$ |
| DeepResearcher + GPT4.1 | $0.115 - 0.212$ | $0.048 - 0.104$ | $0.058 - 0.118$ |

Table 25: Bootstrap 95% Confidence Intervals.

| Model | Precision CI | Recall CI | F1 Score CI |
|---|---|---|---|
| OpenAI DR | 0.552 – 0.701 | 0.442 – 0.587 | 0.465 – 0.607 |
| Perplexity DR | 0.364 – 0.533 | 0.222 – 0.354 | 0.257 – 0.391 |
| Gemini DR | 0.229 – 0.378 | 0.162 – 0.281 | 0.171 – 0.292 |
| DeepResearcher + DS Qwen 32B | 0.123 – 0.257 | 0.037 – 0.105 | 0.052 – 0.126 |
| DeepResearcher + GPT4.1 | 0.123 – 0.260 | 0.040 – 0.119 | 0.053 – 0.135 |

Table 26: Variance across runs (Entities subset).

| Trial | Precision | Recall | F1 |
|---|---|---|---|
| 1 | 0.746 | 0.459 | 0.547 |
| 2 | 0.633 | 0.349 | 0.433 |
| 3 | 0.690 | 0.407 | 0.472 |
| Mean $\pm$ SE | $0.6897 \pm 0.0565$ | $0.4050 \pm 0.0550$ | $0.4840 \pm 0.0579$ |

Table 27: Strict Precision/Recall/F1 scores for OpenAI DR, evaluated with GPT-4o. Std. dev. is across 5 GPT-4o runs (temperature=0.3).

| Subset | Precision | Recall | F1 Score |
|---|---|---|---|
| NovelDS Identification | $0.667 \pm 0.000$ | $0.667 \pm 0.000$ | $0.667 \pm 0.000$ |
| NovelDS Identi-Extraction | $0.450 \pm 0.019$ | $0.388 \pm 0.019$ | $0.406 \pm 0.019$ |
| NovelDS Peer | $0.485 \pm 0.000$ | $0.334 \pm 0.000$ | $0.389 \pm 0.000$ |
| Flights | $0.400 \pm 0.051$ | $0.420 \pm 0.040$ | $0.407 \pm 0.045$ |

Table 28: Strict Precision/Recall/F1 scores for Perplexity DR, evaluated with GPT-4o. Std. dev. is across 5 GPT-4o runs (temperature=0.3).

| Subset | Precision | Recall | F1 Score |
|---|---|---|---|
| NovelDS Identification | $0.500 \pm 0.000$ | $0.500 \pm 0.000$ | $0.500 \pm 0.000$ |
| NovelDS Identi-Extraction | $0.332 \pm 0.015$ | $0.359 \pm 0.017$ | $0.342 \pm 0.016$ |
| NovelDS Peer | $0.318 \pm 0.041$ | $0.126 \pm 0.014$ | $0.174 \pm 0.018$ |
| Flights | $0.173 \pm 0.010$ | $0.176 \pm 0.010$ | $0.174 \pm 0.010$ |

Table 29: Strict Precision/Recall/F1 scores for Gemini DR, evaluated with GPT-4o. Std. dev. is across 5 GPT-4o runs (temperature=0.3).

| Subset | Precision | Recall | F1 Score |
|---|---|---|---|
| NovelDS Identification | $0.333 \pm 0.000$ | $0.333 \pm 0.000$ | $0.333 \pm 0.000$ |
| NovelDS Identi-Extraction | $0.297 \pm 0.037$ | $0.250 \pm 0.023$ | $0.255 \pm 0.027$ |
| NovelDS Peer | $0.295 \pm 0.019$ | $0.148 \pm 0.007$ | $0.196 \pm 0.011$ |
| Flights | $0.154 \pm 0.014$ | $0.157 \pm 0.017$ | $0.156 \pm 0.016$ |

