# OpenReview forum: "Characterizing Deep Research: A Benchmark and Formal Definition"
_ICLR.cc/2026/Conference — ICLR 2026 Poster_

### Official Review · Reviewer_AaxG · 2025-10-30

**Soundness:** 3
**Presentation:** 3
**Contribution:** 2
**Rating:** 4
**Confidence:** 4

**Summary:**

This paper introduces a benchmark for evaluating deep research systems. The authors first formalize deep research type of questions and focused on how it is different from other reasoning tasks. In addition, they evaluates DeepResearch agents on their proposed liveDRbench with 100 questions. Their evaluation is based on accuracy and completeness of claims from ground truth.

**Strengths:**

- The paper makes  make a  formal definition of what constitutes a Deep Research task. I think it is important to take a step to clarify these types of tasks and to distinguish it from multihopQA or long form reasoning.
- The ablation studies and experiments are relatively comprehensive.
- Their use of open web search and “problem inversion” makes the benchmark naturally extensible

**Weaknesses:**

- While the paper claims the benchmark can be easily updated via inversion of long-context tasks, in practice this may be labor-intensive which might require human verification of unique answers and updated ground truth each time the web changes.
- Some important DR-related systems as baselines are missing, such as OpenScholar or LLM+Tool-use agents that combine retrieval with API or browser actions. Including these would make the evaluation more representative of current “agentic” DR methods.
- Measuring coverage over issued queries rather than retrieved gold documents may be misleading. what if different query phrasings can retrieve equivalent document sets. Evaluating retrieval coverage (recall of relevant documents) would be a fairer proxy compared to query coverage.
- If I understood correctly, the claim-level F1 may penalize systems unfairly if entities are represented differently (e.g., “UCLA” vs. “University of California, Los Angeles”). Maybe using some form of soft matching could yield a more realistic measure of factual alignment.
- First, who were the human validators for the 100 questions? If the questions are truly expert-level and niche, it’s not easy for non-experts to validate them. Second, there should be some level of human validation on the evaluation itself to make sure the results are reliable.
- I think there is a chance that because the benchmark construction process uses automated inversion and possibly LLM assistance, some ground-truth claims or links might be hallucinated or incomplete. Human audits could help mitigate this.

**Questions:**

- How practical is the claim that LIVEDRBENCH can be “easily updated”? Given the need for verifying unique answers and refreshing ground truth as the web evolves, what is the actual human effort involved in maintaining the benchmark?
- Could the claim-level F1 metric unfairly penalize systems for surface-level differences in entity names (e.g., “UCLA” vs. “University of California, Los Angeles”)?
- Who validated the 100 benchmark questions? If these questions are specialized and domain-heavy, how were they verified for correctness by non-experts?
-Was there any human validation of the evaluation results to understand to what extend  the automatic scoring is trustworthy?
- Since the benchmark construction involves automatic inversion and possibly LLM-generated ground truth, how do you ensure that no hallucinated or incomplete claims made it into the final dataset?

---

> ### Author Response · Authors · 2025-11-23
> **Rebuttal Part 1**
>
> We sincerely thank you for your efforts in reviewing our manuscript.
>
> > 1. While the paper claims the benchmark can be easily updated via inversion of long-context tasks, in practice this may be labor-intensive which might require human verification of unique answers and updated ground truth each time the web changes.
>
> **Ground Truth expansion.** Thank you for raising this point. In our expansion process candidate answers generated by any system are not accepted automatically but are manually verified against the source documents before being added to the ground-truth set. This process does require human verification. However, the task is strictly verification; checking whether a claim is relevant to the query (e.g., Did the mentioned person win an IMO medal? or Does Zno have a band gap of 3.37eV?). This process is not labor intensive and requires simply checking if the claim is present in the source documents rather than specialized scientific or technical expertise.
>
> > Some important DR-related systems as baselines are missing, such as OpenScholar or LLM+Tool-use agents that combine retrieval with API or browser actions. Including these would make the evaluation more representative of current “agentic” DR methods.
>
> **Baselines.** We do include LLM+tool-use agents as non-DR baselines (see Appendix), so our evaluation already covers that paradigm. Regarding OpenScholar: while it is a retrieval-augmented LM for scientific literature, it uses a static datastore of 45 M open-access papers rather than operating over the open web. This means that OpenScholar won't work at all for Flights and Entities that cover general events or PriorArt and SciFacts that covers a broader scientific literature than their index. Thus, its design and evaluation differ substantially from our open-web, search-intensive deep-research setting. It is, therefore, not a fully agentic DR system in the sense of dynamic web exploration, and this is why it was not included as a baseline.
>
> > Measuring coverage over issued queries rather than retrieved gold documents may be misleading. what if different query phrasings can retrieve equivalent document sets. Evaluating retrieval coverage (recall of relevant documents) would be a fairer proxy compared to query coverage.
>
> **Query-Coverage Analysis.** Thank you for the suggestion. We agree that query-level coverage is only an approximate signal, and we emphasize that our main evaluation (both human and judge-based) measures claim coverage and correctness, not query coverage (Tables 1–3). Since we already use gold documents in our main evaluations, we conduct additional analysis of the "reasoning steps". The query-coverage study is therefore only included as an additional analysis to understand whether DR systems invoke the key information-seeking steps needed to answer multi-claim questions. As described in the paper, we generate necessary queries (both characteristic and extractive) by examining the gold source documents required to support each claim. Coverage is computed as the proportion of semantically matching queries found in a model’s trace, where matching is performed using an LLM judge specifically to account for paraphrases and alternative phrasings. We will clarify that this analysis is complementary and diagnostic, and that the primary benchmark results rely on claim-grounded evaluation.
>
> >If I understood correctly, the claim-level F1 may penalize systems unfairly if entities are represented differently (e.g., “UCLA” vs. “University of California, Los Angeles”). Maybe using some form of soft matching could yield a more realistic measure of factual alignment.
>
> **Claim Level Evaluation.** To clarify, our claim-level F1 is indeed a "soft" matching. It is based on a LLM prompt. We include the following instructions in the eval prompt, as given in our supplementary:
>
> ```
> - If the value is a string, consider them equivalent if they are similar in meaning, or if one is a
> subset of the other.
> - Acronyms or shortened names are equivalent to their expanded forms.
> - ”BERT” and ”BERT: Pre-training of Deep Bidirectional Transformers for Language
> Understanding” −→ equivalent
> - ”ICLR” and ”International Conference on Learning Representations” −→ equivalent
> - ”CNN” and ”ResNet” −→ equivalent (both are convolutional neural networks)
> - ”Adam” and ”SGD” −→ not equivalent (different optimizers)
> - ”0.8” and ”0.80” −→ equivalent numerically
> - Allow for minor differences in formatting, such as casing, whitespace, or common
> abbreviations.
> ```
>
> ```
> - Allow for minor differences in formatting, such as casing, whitespace, or common abbreviations.
> ```
>
> This ensures that the answers are not unfairly penalized.

---

> ### Author Response · Authors · 2025-11-23
> **Rebuttal Part 2**
>
> > First, who were the human validators for the 100 questions? If the questions are truly expert-level and niche, it’s not easy for non-experts to validate them. Second, there should be some level of human validation on the evaluation itself to make sure the results are reliable.
>
> **Human Verification on evaluation.** Three authors participated in the human evaluation. We would like to clarify that the task is strictly verification; checking whether a claim is supported by the cited source. This does not require not domain-expert judgement. This requires simply checking if answer if present in source documents rather than specialized scientific or technical expertise. E.g., checking if ZnO has a band gap of 3.37eV at room temperature is a simple task. We perform human evaluation on the entire LiveDRBench and provide the results in Table 6 of the Appendix. Additionally, we computed the alignment between human and LLM based evaluation and report Cohen's Kappa, MAE in the tables below.
>
>
> Table 1: Cohen Kappa preference scores between Human vs GPT4o
>
> |            Comparison   | Precision | Recall | F1   |
> | ----------------------- | --------- | ------ | ---- |
> | OpenAI vs Perplexity DR | 0.98      | 0.98   | 0.96 |
> | Perplexity vs Gemini DR | 0.86      | 0.92   | 0.86 |
> | Gemini vs OpenAI DR     | 0.98      | 0.96   | 0.96 |
>
> Table 2: MAE between Human vs GPT4o, based on metric on column axis:
>
> |               | Precision | Recall | F1    |
> | ------------- | --------- | ------ | ----- |
> | OpenAI DR     | 0.020     | 0.012  | 0.014 |
> | Perplexity DR | 0.022     | 0.022  | 0.021 |
> | Gemini DR     | 0.035     | 0.027  | 0.028
>
> > I think there is a chance that because the benchmark construction process uses automated inversion and possibly LLM assistance, some ground-truth claims or links might be hallucinated or incomplete. Human audits could help mitigate this.
>
> **Human Verification on evaluation.** Thank you for the comment. We clarify that all ground-truth claims and links were human-audited during dataset construction. Candidate answers produced by any system were never accepted automatically: each was manually checked against the source documents before inclusion. While completeness cannot be guaranteed, this expansion-and-verification process significantly reduces false negatives and ensures that every claim and subclaim in the ground truth is human verified.
>
> > How practical is the claim that LIVEDRBENCH can be “easily updated”? Given the need for verifying unique answers and refreshing ground truth as the web evolves, what is the actual human effort involved in maintaining the benchmark?
>
> **Easy to Update.** Our work is designed such that it can be updated in the following ways. (a) SciFacts questions can easily be updated by using other long-form reasoning benchmarks through our inversion pipeline, (b) NovelDS and PriorArt questions can also be updated with more recent articles, (c) Entities tasks, for example list of IMO winners from a particular country can be extended by including other Olympiads, Movies, etc; and (d) Flights tasks can also be updated with more recent world events. This ensures that LiveDRBench is robust to staleness.
>
> These updates require human verification of new claims, as with the original construction, but the overall process remains lightweight and structured as described above. Human effort is limited to validating the set of unique candidate claims surfaced during expansion.
>
> > Could the claim-level F1 metric unfairly penalize systems for surface-level differences in entity names (e.g., “UCLA” vs. “University of California, Los Angeles”)?
>
> We clarify our soft claim level matching in section **Claim Level Evaluation. ** of the rebuttal. Additionally, we provide the prompt for evaluation in the appendix which exactly addresses this concern.

---

### Official Review · Reviewer_fjWB · 2025-10-31

**Soundness:** 3
**Presentation:** 3
**Contribution:** 3
**Rating:** 6
**Confidence:** 3

**Summary:**

The paper proposes a formalization of “deep research” (DR) as tasks that are both search-intensive (high fan-out over information units) and reasoning-intensive (non-trivial effort to find/process/combine evidence). It argues that the core of DR is claim synthesis rather than long-form prose generation, and introduces LIVEDRBENCH, a 100-task open-web benchmark with a claims-based evaluation (precision/recall over nested claims, with strict scoring that zeros out a claim when any subclaim is wrong). The benchmark spans eight categories (SCIFACTS–Materials/Geo, NOVELDS—three types, PRIORART, ENTITIES, FLIGHTS), and evaluates proprietary DR systems and open-source baselines, reporting best overall performance for OpenAI’s DR model (avg. F1 ≈ 0.55) and analyzing search traces for breadth/depth coverage.

**Strengths:**

Clear problem decomposition. Separating “claims synthesis” from “report generation” is crisp and useful; the DAG framing of queries→evidence→claims aligns with how DR agents operate.

Objective evaluation for multi-claim outputs. The nested claim/subclaim metric operationalizes “grounded completeness” better than stylistic LLM-judge scores used in other DR evaluations.

Useful positioning vs. prior benchmarks. The paper contrasts LIVEDRBENCH with report-quality benchmarks (e.g., DeepResearch Bench) and browsing-centric datasets (e.g., BrowseComp) and motivates claim-level scoring for open-web DR.

**Weaknesses:**

Metric dependence on LLM judging and design choices. Claim agreement and “necessary query” identification rely on LLMs (GPT-4o) and bespoke prompts; the paper needs stronger validation (e.g., human adjudication on a stratified sample, inter-rater agreement, and sensitivity analyses for the “zero credit if any subclaim is wrong” rule).

Reproducibility of proprietary model comparisons. Evaluations for commercial DR systems are done via chat UIs without API parity; differences in browsing tools, guardrails, or hidden budgets could confound scores. More details (and repeated trials) are needed.

Scope overlap with related work could be made sharper. While the paper positions against BrowseComp (short, verifiable answers) and DeepResearchGym (static corpora like ClueWeb22/FineWeb), the narrative could more explicitly articulate complementary use—e.g., LIVEDRBENCH for open-web, claim-rich synthesis; DeepResearchGym for reproducible search sandboxes; DeepResearch Bench for report-quality LLM-judge eval.

**Questions:**

How robust are scores to judge model choice (GPT-4o vs. alternatives) and to noise in claim parsing? Please include a cross-judge concordance table.

For “necessary query coverage,” what fraction was human-validated, and what is agreement between annotators vs. LLM labeling?

How do results change if you relax the strict subclaim rule (e.g., soft weights), and if you weight claims by importance?

---

> ### Author Response · Authors · 2025-11-23
> **Rebuttal Part 1**
>
> We sincerely thank you for your feedback. Below we address each question and concern.
>
> > Metric dependence on LLM judging and design choices. Claim agreement and “necessary query” identification rely on LLMs (GPT-4o) and bespoke prompts; the paper needs stronger validation (e.g., human adjudication on a stratified sample, inter-rater agreement, and sensitivity analyses for the “zero credit if any subclaim is wrong” rule).
>
> Thank you for raising these points. We agree that additional our judge based evaluation would benefit from additional validation. Following for feedback, we conducted the following additional experiments:
>
> (i) **Claim Agreement (Human vs Judge).** To provide additional support for our claims, we perform human evaluation and present our results in Table 6 of the main paper. Three authors participated in this human evaluation. We would like to clarify that the task is strictly verification; checking whether a claim is supported by the cited source. This requires simply checking if the claim is present in the source documents rather than specialized scientific or technical expertise. For example, checking if ZnO has a band gap of 3.37eV at room temperature is significantly easier than finding materials with a band gap of 3.37eV. Therefore, we argue that inter-rater agreement is not required in our evaluation. That said, we agree that validating the LLM judge against human judgment would strengthen our paper. We compared the GPT4o judgement with Human evaluation using (a) Cohen Kappa preference score (following DeepResearchGym), and (b) Mean Absolute Error. Cohen's Kappa (in Table 1 below) measures the agreement between the human evaluators and the judge model against random chance. Concretely, we compare the human evaluation scores in Table 6 (main paper) with GPT4o scores (Table 3 main paper). Three authors performed this human evaluation. A Cohen's Kappa of 1 denotes a perfect agreement and 0 represents disagreement.
>
>
> Table 1: Cohen Kappa preference scores between Human vs GPT4o
>
> |            Comparison   | Precision | Recall | F1   |
> | ----------------------- | --------- | ------ | ---- |
> | OpenAI vs Perplexity DR | 0.98      | 0.98   | 0.96 |
> | Perplexity vs Gemini DR | 0.86      | 0.92   | 0.86 |
> | Gemini vs OpenAI DR     | 0.98      | 0.96   | 0.96 |
>
> Table 2: MAE between Human vs GPT4o, based on metric on column axis:
>
> |               | Precision | Recall | F1    |
> | ------------- | --------- | ------ | ----- |
> | OpenAI DR     | 0.020     | 0.012  | 0.014 |
> | Perplexity DR | 0.022     | 0.022  | 0.021 |
> | Gemini DR     | 0.035     | 0.027  | 0.028 |
>
>
> Our results indicate a very strong agreement between human and judge. This is expected because the task of the judge model is simply to check whether the generated claims match the ground truth claims, which is an easy task.
>
> (ii) **Query-Coverage Analysis.** We agree that query-level coverage is only an approximate signal, and we emphasize that our main evaluation (both human and judge-based) measures claim coverage and correctness, not query coverage (Tables 1–3). Since we already use gold documents in our main evaluations, we conduct additional analysis of the "reasoning steps". The query-coverage study is therefore only included as an additional analysis to understand whether DR systems invoke the key information-seeking steps needed to answer multi-claim questions. We will clarify that this analysis is complementary and diagnostic, and that the primary benchmark results rely on claim-grounded evaluation.

---

> ### Author Response · Authors · 2025-11-23
> **Rebuttal Part 2**
>
> (continued)
> (iii) **Sensitivity Analysis of our metric.** Thank you for your suggestion. Following you recommendation, we conducted evaluations using the strict version of our metric (zero credit if any subclaim is wrong). We report this metric only for the NovelDS and Flights subsets, since these subsets involve multiple subclaims under each main claim. In SciFacts Geo, PriorArt, and Entities, there are only main claims with no subclaims. In SciFacts Materials, there is a single subclaim. In either case, taking a min over the subclaims has no effect on the rankings. We present these alternative evaluation results below. The trend among DR models remains consistent under the strict min metric as well.
>
> Table 11: Strict Precision/Recall/F1 Scores for OpenAI DR, evaluated with GPT4o. Stddev is accross 5 runs of GPT4o, with temperature 0.3.
>
> | Subset                    | Precision     | Recall        | F1 Score      |
> | ------------------------- | ------------- | ------------- | ------------- |
> | NovelDS Identification    | 0.667 ± 0.000 | 0.667 ± 0.000 | 0.667 ± 0.000 |
> | NovelDS Identi-Extraction | 0.450 ± 0.019 | 0.388 ± 0.019 | 0.406 ± 0.019 |
> | NovelDS Peer              | 0.485 ± 0.000 | 0.334 ± 0.000 | 0.389 ± 0.000 |
> | Flights                   | 0.400 ± 0.051 | 0.420 ± 0.040 | 0.407 ± 0.045 |
>
> Table 12: Strict Precision/Recall/F1 Scores for Perplexity DR, evaluated with GPT4o. Stddev is accross 5 runs of GPT4o, with temperature 0.3.
>
> | Subset                    | Precision     | Recall        | F1 Score      |
> | ------------------------- | ------------- | ------------- | ------------- |
> | NovelDS Identification    | 0.500 ± 0.000 | 0.500 ± 0.000 | 0.500 ± 0.000 |
> | NovelDS Identi-Extraction | 0.332 ± 0.015 | 0.359 ± 0.017 | 0.342 ± 0.016 |
> | NovelDS Peer              | 0.318 ± 0.041 | 0.126 ± 0.014 | 0.174 ± 0.018 |
> | Flights                   | 0.173 ± 0.010 | 0.176 ± 0.010 | 0.174 ± 0.010 |
>
> Table 13: Strict Precision/Recall/F1 Scores for Gemini DR, evaluated with GPT4o. Stddev is accross 5 runs of GPT4o, with temperature 0.3.
>
> | Subset                    | Precision     | Recall        | F1 Score      |
> | ------------------------- | ------------- | ------------- | ------------- |
> | NovelDS Identification    | 0.333 ± 0.000 | 0.333 ± 0.000 | 0.333 ± 0.000 |
> | NovelDS Identi-Extraction | 0.297 ± 0.037 | 0.250 ± 0.023 | 0.255 ± 0.027 |
> | NovelDS Peer              | 0.295 ± 0.019 | 0.148 ± 0.007 | 0.196 ± 0.011 |
> | Flights                   | 0.154 ± 0.014 | 0.157 ± 0.017 | 0.156 ± 0.016 |
>
> >Reproducibility of proprietary model comparisons. Evaluations for commercial DR systems are done via chat UIs without API parity; differences in browsing tools, guardrails, or hidden budgets could confound scores. More details (and repeated trials) are needed.
>
> Thank you for raising this point. At the time of writing this paper, proprietary DR models could only be accessed through the UI and their internal budgets were not configurable. To address this concern, we did multiple runs of the DR model after availability of the API. Due to the high cost of running, we do this test on a subset of questions in LiveDRBench on O3-Deep-Research and present our results across multiple evaluation runs in Table 11 below. Since we are unaware about the hyperparameters used in the UI, we follow the recommended prompt generation, default reasoning setting and set max_tool_calls to 500.
>
> Table 11: Varience across runs - Entities
> |   Trail no. | Precision        | Recall           | F1               |
> |-------------|------------------|------------------|------------------|
> |      1      | 0.746            | 0.459            | 0.547            |
> |      2      | 0.633            | 0.349            | 0.433            |
> |      3      | 0.69             | 0.407            | 0.472            |
> |Mean ± StdErr| 0.6897 ± 0.0565  | 0.4050 ± 0.0550  | 0.4840 ± 0.0579  |
>
> > Scope overlap with related work could be made sharper. While the paper positions against BrowseComp (short, verifiable answers) and DeepResearchGym (static corpora like ClueWeb22/FineWeb), the narrative could more explicitly articulate complementary use—e.g., LIVEDRBENCH for open-web, claim-rich synthesis; DeepResearchGym for reproducible search sandboxes; DeepResearch Bench for report-quality LLM-judge eval.
>
> **Clarification on related work.** Thank you for this suggestion. We agree that the relationship to BrowseComp, DeepResearchGym, and DeepResearchBench can be articulated more explicitly. LiveDRBench is designed for open-web, claim-rich deep research tasks where models must find, integrate, and verify multiple pieces of evidence in dynamic settings. In contrast, DeepResearchGym provides a static, reproducible search sandbox focused on controlled retrieval environments, and DeepResearchBench evaluates report-quality using LLM-judge and RAG-based scoring. We will clarify this final version.

---

> ### Author Response · Authors · 2025-11-23
> **Rebuttal Part 3**
>
> > How robust are scores to judge model choice (GPT-4o vs. alternatives) and to noise in claim parsing? Please include a cross-judge concordance table.
>
> Thank you for this suggestion. To further establish the robustness of our evaluation framework, we perform additional evalutions with Qwen32B. Further,  in Tables 3, 4, 5 (response to reviewer vd99) we run our llm-as-judge model GPT4o multiple times and show that the variance across multiple runs is $\leq$ 0.02.
>
> Table 6: Stddev accross 5 Qwen3-32B Non-Thinking eval runs for OpenAI DR, with temperature 0.3
>
> | Subset                    | Precision     | Recall        | F1            |
> | ------------------------- | ------------- | ------------- | ------------- |
> | SciFacts Materials        | 0.332 ± 0.028 | 0.355 ± 0.055 | 0.343 ± 0.000 |
> | SciFacts Geo              | 0.715 ± 0.000 | 0.728 ± 0.000 | 0.721 ± 0.000 |
> | NovelDS Identification    | 0.667 ± 0.000 | 0.667 ± 0.000 | 0.667 ± 0.000 |
> | NovelDS Identi-Extraction | 0.506 ± 0.009 | 0.415 ± 0.008 | 0.438 ± 0.009 |
> | NovelDS Peer              | 0.789 ± 0.008 | 0.476 ± 0.003 | 0.580 ± 0.005 |
> | PriorArt                  | 0.714 ± 0.000 | 0.473 ± 0.000 | 0.552 ± 0.000 |
> | Entities                  | 0.811 ± 0.000 | 0.532 ± 0.000 | 0.600 ± 0.000 |
> | Flights                   | 0.589 ± 0.023 | 0.572 ± 0.016 | 0.574 ± 0.019 |
> | Overall Avg               | 0.640 ± 0.008 | 0.527 ± 0.010 | 0.559 ± 0.004 |
>
> Table 7: Stddev accross 5 Qwen3-32B Non-Thinking eval runs for Perplexity DR, with temperature 0.3
>
> | Subset                    | Precision     | Recall        | F1            |
> | ------------------------- | ------------- | ------------- | ------------- |
> | SciFacts Materials        | 0.264 ± 0.000 | 0.164 ± 0.000 | 0.202 ± 0.000 |
> | SciFacts Geo              | 0.363 ± 0.011 | 0.233 ± 0.036 | 0.284 ± 0.036 |
> | NovelDS Identification    | 0.633 ± 0.000 | 0.633 ± 0.000 | 0.633 ± 0.000 |
> | NovelDS Identi-Extraction | 0.358 ± 0.016 | 0.385 ± 0.018 | 0.369 ± 0.016 |
> | NovelDS Peer              | 0.565 ± 0.008 | 0.264 ± 0.003 | 0.337 ± 0.004 |
> | PriorArt                  | 0.704 ± 0.006 | 0.351 ± 0.004 | 0.432 ± 0.005 |
> | Entities                  | 0.666 ± 0.000 | 0.351 ± 0.000 | 0.431 ± 0.000 |
> | Flights                   | 0.447 ± 0.042 | 0.381 ± 0.036 | 0.405 ± 0.038 |
> | Overall Avg               | 0.500 ± 0.010 | 0.345 ± 0.012 | 0.387 ± 0.012 |
>
> Table 8: Stddev accross 5 Qwen3-32B Non-Thinking eval runs for Gemini DR, with temperature 0.3
>
> | Subset                    | Precision     | Recall        | F1            |
> | ------------------------- | ------------- | ------------- | ------------- |
> | SciFacts Materials        | 0.075 ± 0.003 | 0.013 ± 0.000 | 0.022 ± 0.001 |
> | SciFacts Geo              | 0.405 ± 0.000 | 0.272 ± 0.005 | 0.326 ± 0.005 |
> | NovelDS Identification    | 0.400 ± 0.000 | 0.400 ± 0.000 | 0.400 ± 0.000 |
> | NovelDS Identi-Extraction | 0.397 ± 0.016 | 0.306 ± 0.019 | 0.333 ± 0.016 |
> | NovelDS Peer              | 0.552 ± 0.006 | 0.250 ± 0.002 | 0.339 ± 0.003 |
> | PriorArt                  | 0.181 ± 0.000 | 0.147 ± 0.000 | 0.141 ± 0.000 |
> | Entities                  | 0.451 ± 0.001 | 0.307 ± 0.001 | 0.342 ± 0.001 |
> | Flights                   | 0.280 ± 0.023 | 0.314 ± 0.030 | 0.295 ± 0.026 |
> | Overall Avg               | 0.343 ± 0.006 | 0.251 ± 0.007 | 0.275 ± 0.007 |
>
> > For “necessary query coverage,” what fraction was human-validated, and what is agreement between annotators vs. LLM labeling?
>
> We computed MAE and Cohen's Kappa to measure the alignment between human annotators and LLM judge (in Table 1, 2). The alignment between humans and LLM judge for all our models is very high -- 92.67% Cohen's F1 Score on average. All our examples were human validated and the results are presented in Table 6 of the main paper.
>
> >  How do results change if you relax the strict subclaim rule (e.g., soft weights), and if you weight claims by importance?
>
> Tables 10, 11, and 12 in the rebuttal show that changing the default metric to the strict scoring variant (which assigns zero credit to a claim if any of its subclaims are incorrect) leaves the rankings effectively unchanged across all datasets. This indicates that our results are robust to tightening the scoring rule.
>
> Regarding weighting claims by importance: assigning importance weights is inherently subjective and depends on downstream application goals. To keep the benchmark well-defined and evaluation-neutral, we deliberately evaluate only the two extremes, uniform weighting and the fully strict variant. Exploring importance-weighted scoring is an interesting direction but outside the scope of this work. We will incorporate the strict-scoring results into the revised manuscript for clarity.

---

### Official Review · Reviewer_5YJ8 · 2025-11-01

**Soundness:** 2
**Presentation:** 2
**Contribution:** 3
**Rating:** 4
**Confidence:** 4

**Summary:**

The paper proposes a formal definition of *deep research* (DR) as the sub‑task of claim synthesis over a large number of information units, and introduces **LiveDRBench**, a publicly released benchmark of 100 web‑search‑driven queries spanning scientific, dataset‑finding, prior‑art, entity‑listing and flight‑incident domains.  The benchmark is built by “problem inversion” of existing long‑context QA items and is evaluated through a claim‑based precision/recall metric that is computed by an LLM judge (GPT‑4o).  Using this infrastructure the authors compare three proprietary DR agents (OpenAI, Perplexity, Gemini) and two open‑source agents, reporting average F1 scores ranging from 0.02 to 0.72 and analysing trace‑level search coverage, depth, branching and back‑tracking.

The paper tackles an important, under‑explored problem and provides a promising benchmark.  However, the **evaluation methodology relies heavily on an opaque LLM judge**, and suffers from reproducibility issues (especially for proprietary agents).  These methodological shortcomings significantly limit confidence in the reported claims and hinder the benchmark’s immediate usefulness to the community. Please see my comments below.

**Strengths:**

- This is a timely and relevant problem: the framing also separates information synthesis (claims) from long-form report generation, improving construct clarity for DR evaluation.
- Claim-based evaluation design directly targets correctness/completeness of substantive content rather than stylistic quality; strict metrics are well-motivated for enumeration tasks.
- LiveDRBench spans multiple realistic DR use cases, including science and public-interest scenarios; explicit effort to make benchmark updatable is valuable.
- Transparent reporting of category-wise weakness (e.g., SCIFACTS Materials) and plausible sources of difficulty (subclaim extraction across long documents).
- Provides detailed prompts and evaluation pipeline in appendices; attempts to validate LLM-judging by a human pass that yields “similar” aggregates.
- Trace-analysis perspective (breadth/depth/coverage) is practically insightful and aligns with qualitative failure modes of DR agents.

**Weaknesses:**

- Proprietary DR systems were evaluated via chat UIs with unspecified/variable budgets; non-DR baselines used APIs with fixed “≈30s” reasoning but DR systems seemingly ran with much larger, uncontrolled search/time budgets. This undermines comparability and may inflate proprietary DR performance relative to baselines and open-source agents.
- Single-run reporting without variance: no per-task variance, confidence intervals, or statistical tests. Given high run-to-run variability in browsing agents and changing web results, this is a major limitation for drawing strong comparative claims.
- Ephemeral open-web setup without timestamped corpora or cached sources; no normalization for geographic or temporal search variability. The “Live” design is appealing but compromises replicability without web snapshots or logged artifacts.
- LLM-as-judge dependence: GPT-4o is used for matching, scoring, and even for trace analyses (necessary queries, dependency classification, branching/backtracking). Although authors include a “human” table with similar means, there is no inter-annotator agreement, correlation coefficients, or systematic error analysis. Heavy reliance on a single proprietary judge risks evaluation bias.
- Ground-truth expansion procedure: the rubric is augmented with any valid answers produced by the evaluated systems. While pragmatic, this can introduce dependence of the label space on model outputs, advantaging models that are included (and potentially those aligned with the judge). It also does not guarantee completeness as unseen correct answers remain out-of-scope, distorting recall.
- Metric design choices not stress-tested: the multiplicative/product formulation across subclaims can collapse scores when one subclaim is wrong; no ablations comparing product vs min (strict) vs additive/averaged schemes and their impact on ranking stability. Weighting scheme w_i is suggested but not analyzed.
- Claims that OpenAI DR “performs best” are plausible but not sufficiently supported for a benchmark paper because (a) budgets and interaction settings are not matched, (b) no statistical testing, (c) potential judge/model coupling (OpenAI DR and GPT-4o judge), and (d) evaluation performed on live web without controls.
- Trace-analysis conclusions (e.g., “coverage ~ half”) rely on GPT-4o-generated “necessary queries” and GPT-4o matching in traces, with no human validation or inter-rater reliability. As such, these are best viewed as exploratory rather than definitive evidence of breadth/depth gaps.

**Questions:**

1. Fairness/budget control: How many tokens, queries, and total wall-clock time did each system consume per task? Can you re-run proprietary DR systems with matched query/time budgets to non-DR baselines (and to each other) via APIs?
2. Variance and significance: Can you report per-task distributions, bootstrap CIs, and statistical tests for pairwise comparisons? How stable are rankings across multiple runs?
3. LLM-judge reliability: What is the agreement between GPT-4o and multiple independent human annotators (e.g., κ or Spearman/Pearson correlations per category)? Any cross-judge validation (e.g., Claude/LLama judges)?
4. Ground-truth expansion: How many items were added via model outputs per category? Did any system preferentially contribute to expansions? Can you freeze a versioned ground-truth independent of the measured systems to avoid circularity?
5. Replicability: Will you release time-stamped queries, full model traces, and archival snapshots of cited sources (e.g., WARC) so results can be reproduced?
6. Metrics sensitivity: How robust are system rankings to alternative claim aggregation strategies (product vs mean vs strict-min) and claim weighting? Please include rank correlations across metric variants.
7. Human evaluation details: How many human annotators participated in the “human‑evaluated” verification, what were their expertise levels, and what were the agreement statistics?
8. Open‑source agent failures: Could you provide logs or error analyses for the DeepResearcher runs that “infinite‑looped” or “failed”?  Might alternative retrieval back‑ends or hyper‑parameter tuning change their performance?

---

> ### Author Response · Authors · 2025-11-23
> **Rebuttal Part 1**
>
> We sincerely thank you for your time and effort in reviewing our manuscript.
>
> **Proprietary DR and non-DR baselines.** Thank you for your comment. At the time of writing this paper, proprietary DR models could only be accessed through the UI and their internal budgets were not configurable. Importantly, our main goal was to showcase the importance of reasoning in achieving higher scores for our benchmark. Following this recommendation, we ran experiments on non-DR models with reasoning set to high, with and without improved prompts(Table 9). Our results further support our claims as the model's performance improves substantially with improved reasoning.
>
> Table 9: Non-DR models with high reasoning
>
> | Model                         | Precision | Recall | F1    |
> |-------------------------------|-----------|--------|-------|
> | o3 with efficient prompt      | 0.8       | 0.616  | 0.666 |
> | o3                            | 0.645     | 0.492  | 0.539 |
> | o4_mini with efficient prompt | 0.72      | 0.341  | 0.402 |
>
> **Variance, Statistical tests.** Thank you for your suggestion. To address your concern, we conducted multple additional experiments -
>
> (i) We performed statistical tests to measure the alignment between llm-as-judge scores and human evaluation, showing a high Cohen's Kappa score and a low MAE. The results are presented in Table 1 and 2 of the rebuttal.
>
> Table 1: Cohen Kappa preference scores between Human vs GPT4o
>
> |            Comparison   | Precision | Recall | F1   |
> | ----------------------- | --------- | ------ | ---- |
> | OpenAI vs Perplexity DR | 0.98      | 0.98   | 0.96 |
> | Perplexity vs Gemini DR | 0.86      | 0.92   | 0.86 |
> | Gemini vs OpenAI DR     | 0.98      | 0.96   | 0.96 |
>
> Table 2: MAE between Human vs GPT4o, based on metric on column axis:
>
> |               | Precision | Recall | F1    |
> | ------------- | --------- | ------ | ----- |
> | OpenAI DR     | 0.020     | 0.012  | 0.014 |
> | Perplexity DR | 0.022     | 0.022  | 0.021 |
> | Gemini DR     | 0.035     | 0.027  | 0.028 |
>
> (ii) We computed per-example standard error in the evaluation of the DR models on LiveDRBench and report our results in Table 10 below. Our observations indicate a low standard error across all five models, validating the robustness and the difficulty of our benchmark. We will add these results to the revised manuscript.
>
> Table 10: Per example variability
>
> | Model                        | Precision ± StdErr | Recall ± StdErr | F1 Score ± StdErr |
> | ---------------------------- | ------------------ | --------------- | ----------------- |
> | OpenAI DR                    | 0.633 ± 0.031      | 0.523 ± 0.034   | 0.555 ± 0.032     |
> | Perplexity DR                | 0.466 ± 0.037      | 0.317 ± 0.029   | 0.355 ± 0.030     |
> | Gemini DR                    | 0.335 ± 0.031      | 0.240 ± 0.025   | 0.263 ± 0.025     |
> | DeepResearcher + DS Qwen 32B | 0.166 ± 0.024      | 0.069 ± 0.013   | 0.086 ± 0.015     |
> | DeepResearcher + GPT4.1      | 0.163 ± 0.025      | 0.076 ± 0.014   | 0.088 ± 0.015     |
>
> We present Confidence Intervals (95%) for Precision, Recall, and F1 Score evaluation of the DR models on LiveDRBench. Table 14 shows the confidence intervals assuming a Gaussian distribution. To avoid the Gaussian assumption, we also compute bootstrap confidence intervals as shown in Table 15. (see Rebuttal Part 6)
>
> (iii) We did multiple runs of the DR model after availability of the API. Due to the high cost of running, we do this test on a subset of questions in LiveDRBench on O3-Deep-Research and present our results across multiple evaluation runs in Table 11 below. Since we are unaware about the hyperparameters used in the UI, we follow the recommended prompt generation, default reasoning setting and set max_tool_calls to 500.
>
> Table 11: Varience across runs - Entities
> |   Trail no. | Precision        | Recall           | F1               |
> |-------------|------------------|------------------|------------------|
> |      1      | 0.746            | 0.459            | 0.547            |
> |      2      | 0.633            | 0.349            | 0.433            |
> |      3      | 0.69             | 0.407            | 0.472            |
> |Mean ± StdErr| 0.6897 ± 0.0565  | 0.4050 ± 0.0550  | 0.4840 ± 0.0579  |

---

> ### Author Response · Authors · 2025-11-23
> **Rebuttal Part 2**
>
> **Open Web Setting.** Thank you for your comment. We acknowledge that using the open web introduces a temporal and geographical variability. However, this is an intentional choice, LiveDRBench aims to evaluate deep-research capabilities as in deployed real world settings where corpora is constantly evolving. This is essential for measuring the search and reasoning intensity under realistic conditions. To support replicability, we will make our entire dataset, evaluation pipeline public. Since our evaluation depends on fixed ground truth evidence, comparisons across models typically remain stable. Overall, we argue that the live design is a natural tradeoff that sacrifies strict determinism while maintaining reproducibility through fixed ground truth and consistent evaluation pipeline. For instance, OpenAI released BrowseComp, a realistic hard needle-in-the-haystack DR benchmark, and a group of researchers released BrowseComp-Plus, with a fixed index. We believe both kinds of benchmarks have value; the ones without any index especially for evaluating deep research capabilities across the entire internet. To ensure reproducibility, we will release everything necessary to reproduce the evaluation, including code, prompts, datasets, evaluation scripts, and timestamped model traces.
>
> **Ground Truth expansion.** Thank you for raising this point. We augment the ground truth using all the candidate DR models, ensuring that the label space is not biased towards any one model. In our expansion process candidate answers generated by any system are not accepted automatically but are manually verified against the source documents before being added to the ground-truth set. This ensures that the label space does not depend on any one model outputs per se, but only on human-confirmed valid evidence from all DR models. Once verified, expanded answers are added to the ground-truth set and then fixed for all models, ensuring that no system benefits from its own outputs during evaluation. Although not guaranteed to be complete, this expansion strategy reduces false negatives in deep research evaluation while keeping the label space model-agnostic and document-grounded.
>
>
> **Metric design ablation.** Thank you for your feedback. Following you recommendation, we conducted evaluations using the strict version of our metric. We report this metric only for the NovelDS and Flights subsets, since these subsets involve multiple subclaims under each main claim. In SciFacts Geo, PriorArt, and Entities, there are only main claims with no subclaims. In SciFacts Materials, there is a single subclaim. In either case, taking a min over the subclaims has no effect on the evaluation. We present these alternative evaluation results below. The trend among DR models remains consistent under the strict min metric as well.
>
> Table 11: Strict Precision/Recall/F1 Scores for OpenAI DR, evaluated with GPT4o. Stddev is accross 5 runs of GPT4o, with temperature 0.3.
>
> | Subset                    | Precision     | Recall        | F1 Score      |
> | ------------------------- | ------------- | ------------- | ------------- |
> | NovelDS Identification    | 0.667 ± 0.000 | 0.667 ± 0.000 | 0.667 ± 0.000 |
> | NovelDS Identi-Extraction | 0.450 ± 0.019 | 0.388 ± 0.019 | 0.406 ± 0.019 |
> | NovelDS Peer              | 0.485 ± 0.000 | 0.334 ± 0.000 | 0.389 ± 0.000 |
> | Flights                   | 0.400 ± 0.051 | 0.420 ± 0.040 | 0.407 ± 0.045 |
>
> Table 12: Strict Precision/Recall/F1 Scores for Perplexity DR, evaluated with GPT4o. Stddev is accross 5 runs of GPT4o, with temperature 0.3.
>
> | Subset                    | Precision     | Recall        | F1 Score      |
> | ------------------------- | ------------- | ------------- | ------------- |
> | NovelDS Identification    | 0.500 ± 0.000 | 0.500 ± 0.000 | 0.500 ± 0.000 |
> | NovelDS Identi-Extraction | 0.332 ± 0.015 | 0.359 ± 0.017 | 0.342 ± 0.016 |
> | NovelDS Peer              | 0.318 ± 0.041 | 0.126 ± 0.014 | 0.174 ± 0.018 |
> | Flights                   | 0.173 ± 0.010 | 0.176 ± 0.010 | 0.174 ± 0.010 |
>
> Table 13: Strict Precision/Recall/F1 Scores for Gemini DR, evaluated with GPT4o. Stddev is accross 5 runs of GPT4o, with temperature 0.3.
>
> | Subset                    | Precision     | Recall        | F1 Score      |
> | ------------------------- | ------------- | ------------- | ------------- |
> | NovelDS Identification    | 0.333 ± 0.000 | 0.333 ± 0.000 | 0.333 ± 0.000 |
> | NovelDS Identi-Extraction | 0.297 ± 0.037 | 0.250 ± 0.023 | 0.255 ± 0.027 |
> | NovelDS Peer              | 0.295 ± 0.019 | 0.148 ± 0.007 | 0.196 ± 0.011 |
> | Flights                   | 0.154 ± 0.014 | 0.157 ± 0.017 | 0.156 ± 0.016 |

---

> ### Author Response · Authors · 2025-11-23
> **Rebuttal Part 3**
>
> **LLM Judge robustness.** Following your feedback, we conducted additional experiments to validate the robustness of llm-judge model -
>
> (i) We compute the alignment between LLM-judge and human evaluation following DeepResearchGym and present our results in Tables 1, 2 below. Our results indicate a very high alignment between human and llm-judge across the three DR models. This is high agreement is expected because the task of the judge models is simple -- verifying if the generated claims match the ground truth. Cohen's Kappa (in Table 1 below) measures the agreement between the human evaluators and the judge model against random chance. Concretely, we compare the human evaluation scores in Table 6 (main paper) with GPT4o scores (Table 3 main paper). Three authors performed this human evaluation. A Cohen's Kappa of 1 denotes a perfect agreement and 0 represents disagreement.
>
> Table 1: Cohen Kappa preference scores between Human vs GPT4o
>
> |            Comparison   | Precision | Recall | F1   |
> | ----------------------- | --------- | ------ | ---- |
> | OpenAI vs Perplexity DR | 0.98      | 0.98   | 0.96 |
> | Perplexity vs Gemini DR | 0.86      | 0.92   | 0.86 |
> | Gemini vs OpenAI DR     | 0.98      | 0.96   | 0.96 |
>
> Table 2: MAE between Human vs GPT4o, based on metric on column axis:
>
> |               | Precision | Recall | F1    |
> | ------------- | --------- | ------ | ----- |
> | OpenAI DR     | 0.020     | 0.012  | 0.014 |
> | Perplexity DR | 0.022     | 0.022  | 0.021 |
> | Gemini DR     | 0.035     | 0.027  | 0.028 |
>
>
> (ii) We conduct evaluations using an additional judge model Qwen32B and present the results in the Tables below. We find that Qwen3-32B and our existing evaluation with GPT4o are very similar. This is expected because of the simplicity of the evaluation task.
>
> Table 6: Stddev accross 5 Qwen3-32B Non-Thinking eval runs for OpenAI DR, with temperature 0.3
>
> | Subset                    | Precision     | Recall        | F1            |
> | ------------------------- | ------------- | ------------- | ------------- |
> | SciFacts Materials        | 0.332 ± 0.028 | 0.355 ± 0.055 | 0.343 ± 0.000 |
> | SciFacts Geo              | 0.715 ± 0.000 | 0.728 ± 0.000 | 0.721 ± 0.000 |
> | NovelDS Identification    | 0.667 ± 0.000 | 0.667 ± 0.000 | 0.667 ± 0.000 |
> | NovelDS Identi-Extraction | 0.506 ± 0.009 | 0.415 ± 0.008 | 0.438 ± 0.009 |
> | NovelDS Peer              | 0.789 ± 0.008 | 0.476 ± 0.003 | 0.580 ± 0.005 |
> | PriorArt                  | 0.714 ± 0.000 | 0.473 ± 0.000 | 0.552 ± 0.000 |
> | Entities                  | 0.811 ± 0.000 | 0.532 ± 0.000 | 0.600 ± 0.000 |
> | Flights                   | 0.589 ± 0.023 | 0.572 ± 0.016 | 0.574 ± 0.019 |
> | Overall Avg               | 0.640 ± 0.008 | 0.527 ± 0.010 | 0.559 ± 0.004 |
>
> Table 7: Stddev accross 5 Qwen3-32B Non-Thinking eval runs for Perplexity DR, with temperature 0.3
>
> | Subset                    | Precision     | Recall        | F1            |
> | ------------------------- | ------------- | ------------- | ------------- |
> | SciFacts Materials        | 0.264 ± 0.000 | 0.164 ± 0.000 | 0.202 ± 0.000 |
> | SciFacts Geo              | 0.363 ± 0.011 | 0.233 ± 0.036 | 0.284 ± 0.036 |
> | NovelDS Identification    | 0.633 ± 0.000 | 0.633 ± 0.000 | 0.633 ± 0.000 |
> | NovelDS Identi-Extraction | 0.358 ± 0.016 | 0.385 ± 0.018 | 0.369 ± 0.016 |
> | NovelDS Peer              | 0.565 ± 0.008 | 0.264 ± 0.003 | 0.337 ± 0.004 |
> | PriorArt                  | 0.704 ± 0.006 | 0.351 ± 0.004 | 0.432 ± 0.005 |
> | Entities                  | 0.666 ± 0.000 | 0.351 ± 0.000 | 0.431 ± 0.000 |
> | Flights                   | 0.447 ± 0.042 | 0.381 ± 0.036 | 0.405 ± 0.038 |
> | Overall Avg               | 0.500 ± 0.010 | 0.345 ± 0.012 | 0.387 ± 0.012 |
>
> Table 8: Stddev accross 5 Qwen3-32B Non-Thinking eval runs for Gemini DR, with temperature 0.3
>
> | Subset                    | Precision     | Recall        | F1            |
> | ------------------------- | ------------- | ------------- | ------------- |
> | SciFacts Materials        | 0.075 ± 0.003 | 0.013 ± 0.000 | 0.022 ± 0.001 |
> | SciFacts Geo              | 0.405 ± 0.000 | 0.272 ± 0.005 | 0.326 ± 0.005 |
> | NovelDS Identification    | 0.400 ± 0.000 | 0.400 ± 0.000 | 0.400 ± 0.000 |
> | NovelDS Identi-Extraction | 0.397 ± 0.016 | 0.306 ± 0.019 | 0.333 ± 0.016 |
> | NovelDS Peer              | 0.552 ± 0.006 | 0.250 ± 0.002 | 0.339 ± 0.003 |
> | PriorArt                  | 0.181 ± 0.000 | 0.147 ± 0.000 | 0.141 ± 0.000 |
> | Entities                  | 0.451 ± 0.001 | 0.307 ± 0.001 | 0.342 ± 0.001 |
> | Flights                   | 0.280 ± 0.023 | 0.314 ± 0.030 | 0.295 ± 0.026 |
> | Overall Avg               | 0.343 ± 0.006 | 0.251 ± 0.007 | 0.275 ± 0.007 |

---

> ### Author Response · Authors · 2025-11-23
> **Rebuttal Part 4**
>
> **Claims about OpenAI DR Performance.** Thank you for the detailed feedback.
>
> (a) For proprietary DR systems, matching budgets is not feasible because they only expose chat-style interfaces without controllable search or time parameters. Importantly, our goal is not to conduct a controlled compute-fairness benchmark between DR and non-DR agents, rather to show how performance varies across high and low reasoning systems. Non-DR baselines were likewise run with their default recommended settings that was computationally viable, and they were not disadvantaged by any imposed constraints. Based on your feedback, we will avoid highlighting "that openAI DR performs the best" and instead, highlight the overall low accuracy of results across models and settings, that supports the suitability of the benchmark for improving DR.
>
> (b) We do statistical tests to validate the robustness of our llm-judge evaluation. Following DeepResearchGym, we compute Cohen's Kappa to measure the agreement scores and also compute the MAE between human and judge based evaluation.
>
> Table 1: Cohen Kappa preference scores between Human vs GPT4o.
>
> |            Comparison   | Precision | Recall | F1   |
> | ----------------------- | --------- | ------ | ---- |
> | OpenAI vs Perplexity DR | 0.98      | 0.98   | 0.96 |
> | Perplexity vs Gemini DR | 0.86      | 0.92   | 0.86 |
> | Gemini vs OpenAI DR     | 0.98      | 0.96   | 0.96 |
>
> Table 2: MAE between Human vs GPT4o, based on metric on column axis:
>
> |               | Precision | Recall | F1    |
> | ------------- | --------- | ------ | ----- |
> | OpenAI DR     | 0.020     | 0.012  | 0.014 |
> | Perplexity DR | 0.022     | 0.022  | 0.021 |
> | Gemini DR     | 0.035     | 0.027  | 0.028 |
>
>
> (c.) We conduct additional evaluations with Qwen3-32B judge. The results of our additional evaluation closely match our initial Tables, which confirms that there is no biases introduced in our evaluation. The results are presented in Tables 6, 7, 8.
>
> (d) Thank you for your comment. We address this in the answer to your previous concern in **Open Web Setting**.
>
> **Trace Analysis.** Thank you for pointing this out. We agree that our trace-analysis results, which rely on GPT-4o-generated queries and automated matching, should be interpreted as exploratory rather than definitive. We will clarify this limitation in the final version and position the analysis as initial evidence rather than conclusive argument.
>
>
> > Will you release time-stamped queries, full model traces, and archival snapshots of cited sources (e.g., WARC) so results can be reproduced?
>
> **Replicability.** We will release everything necessary to reproduce the evaluation, including code, prompts, datasets, evaluation scripts, and timestamped model traces. For openai dr, which does not provide the exact URL in the trace, just the domains, we will share the domains along with timestamp. We believe this provides strong practical replicability while remaining faithful to the benchmark’s purpose as a live deep-research evaluation.
>
> >  Metrics sensitivity: How robust are system rankings to alternative claim aggregation strategies (product vs mean vs strict-min) and claim weighting? Please include rank correlations across metric variants.
>
> **Strict Metrics.** We report the strict-min metric in Tables 10-12, with the ordering between the proprietary DR models maintained. We report this metric only for the NovelDS and Flights subsets, since these subsets involve multiple subclaims under each main claim. In SciFacts Geo, PriorArt, and Entities, there are only main claims with no subclaims. In SciFacts Materials, there is a single subclaim. In either case, taking a min over the subclaims has no effect on the rankings.
>
> **Product vs Mean metrics.** We use a product-based metric to avoid giving credit for correct subclaims when the main claim is wrong. For example, one benchmark question asks for a specific non-fatal flight incident and its casualty and serious-injury counts (both zero for the correct incident). All proprietary DR models retrieved the wrong non-fatal flight, producing the wrong flight number (main claim) but the right casualty and injury counts (subclaims) simply because the retrieved incident also had zero fatalities. A mean-based metric would incorrectly reward these subclaims; the product correctly assigns zero credit.

---

> ### Author Response · Authors · 2025-11-23
> **Rebuttal Part 5**
>
> > Ground-truth expansion: How many items were added via model outputs per category? Did any system preferentially contribute to expansions? Can you freeze a versioned ground-truth independent of the measured systems to avoid circularity?
>
> Thank for raising this point. We augment the ground truth using all the candidate DR models, ensuring that the ground to is not biased towards any one model's outputs. On an average, we had to use this expansion process approximately 20% of the time.
>
> > Human evaluation details: How many human annotators participated in the “human‑evaluated” verification, what were their expertise levels, and what were the agreement statistics?
>
> Thank you for your question. Three people participated in the human evaluation. We would like to clarify that the task is strictly verification; checking whether a claim is supported by the cited source. This does not require not domain-expert judgement. This requires simply checking if the claim is present in the source documents rather than specialized scientific or technical expertise. For example, checking if ZnO has a band gap of 3.37eV at room temperature is significantly easier than finding materials with a band gap of 3.37eV. We will add the number of annotators and clarify that agreement statistics are less central in this setting.
>
> > Open‑source agent failures: Could you provide logs or error analyses for the DeepResearcher runs that “infinite‑looped” or “failed”? Might alternative retrieval back‑ends or hyper‑parameter tuning change their performance?
>
>  We use default settings of all open source DR agents and use the same retriever back-ends (open-web). This ensures a fair evaluation of performance on our benchmark. We would like to clarify that the HuggingFace Open DR models run into an infinite looped (Lines 412-414). We provide an example reasoning trace, shortened for readability. For brevity, we show the summarized trace from an even simpler question that we initially tested, "What's the stock price of Microsoft 1 month ago?" asked on April 3 2025.
> ```text=
> ---- Step 1 ----
> - Code executed to compute the date “1 month ago”.
> - Output: **2025-03-03**.
>
> ---- Step 2 ----
> - search_agent called with: “Find the stock price of Microsoft on 2025-03-03.”
> - Search results return **only future price forecasts**, not historical data.
> - Model chooses CoinCodex forecast ($389.43 on May 1, 2025) as “closest”.
>
> Then the system tries to produce a final_answer tool call.
>
> ---- Beginning of repetition ----
>
> From Step 2 onward, the trace enters a **loop**.
> Each following step has the same structure:
>
> ----  Step N (3 through ~10) ----
> 1. **Model attempts the exact same tool call**:
>
>
> {"name": "final_answer",
> "arguments": {
> "answer": "### 1. Task outcome (short version):
> The predicted Microsoft stock price on 2025-03-03 is approximately $389.43...
> ### 2. Task outcome (extremely detailed version):
> According to a forecast from CoinCodex...
> ### 3. Additional context (if relevant):
> ..."
> }
> }
>
>
> 2. **The system rejects it with the same JSON parsing error** each time:
>
> Error: Expecting ',' delimiter...
>
> 3. The system prints:
> - "Error while generating or parsing output"
> - The same invalid JSON blob
> - The same explanation
> - A time/tokens summary for the step
>
> 4. The system then moves to **Step N+1**,
> and the model tries **the same invalid JSON tool call again**, causing **the same error again**.
>
> This repeats for:
> - Step 3
> - Step 4
> - Step 5
> - Step 6
> - Step 7
> - Step 8
> - Step 9
> - Step 10
>
> Every iteration:
> - No new search
> - No new reasoning
> - No variation in output
> - No change in the attempted JSON
> - Identical failure
>
> ---- End of repetition ----
>
> ---- Final block ----
> - After the repeated failures, the final block shows no successful resolution.
>
> ```
>
>
> We will add more qualitative examples to the revised version of the manuscript.

---

> ### Author Response · Authors · 2025-11-23
> **Rebuttal Part 6**
>
> Table 14: Confidence Intervals (95%) - Gaussian Assumption
>
> | Model                        | Precision CI  | Recall CI     | F1 Score CI   |
> | ---------------------------- | ------------- | ------------- | ------------- |
> | OpenAI DR                    | 0.571 - 0.695 | 0.456 - 0.588 | 0.484 - 0.610 |
> | Perplexity DR                | 0.393 - 0.538 | 0.253 - 0.368 | 0.283 - 0.401 |
> | Gemini DR                    | 0.277 - 0.399 | 0.191 - 0.288 | 0.208 - 0.306 |
> | DeepResearcher + DS Qwen 32B | 0.118 - 0.214 | 0.044 - 0.095 | 0.057 - 0.115 |
> | DeepResearcher + GPT4.1      | 0.115 - 0.212 | 0.048 - 0.104 | 0.058 - 0.118 |
>
> Table 15: Bootstrap Confidence Intervals (95%)
>
> | Model                        | Precision CI  | Recall CI     | F1 Score CI   |
> | ---------------------------- | ------------- | ------------- | ------------- |
> | OpenAI DR                    | 0.552 - 0.701 | 0.442 - 0.587 | 0.465 - 0.607 |
> | Perplexity DR                | 0.364 - 0.533 | 0.222 - 0.354 | 0.257 - 0.391 |
> | Gemini DR                    | 0.229 - 0.378 | 0.162 - 0.281 | 0.171 - 0.292 |
> | DeepResearcher + DS Qwen 32B | 0.123 - 0.257 | 0.037 - 0.105 | 0.052 - 0.126 |
> | DeepResearcher + GPT4.1      | 0.123 - 0.260 | 0.040 - 0.119 | 0.053 - 0.135 |

---

> > ### Comment · Reviewer_5YJ8 · 2025-11-27
> > **Thanks for addressing my queries; I've raised my score to 6**
> >
> > Dear authors,
> >
> > Thank you for answering my questions in detail. I have gone through them and am satisfied with your responses. As such, I have raised my score to 6.

---

### Official Review · Reviewer_vd99 · 2025-11-09

**Soundness:** 3
**Presentation:** 3
**Contribution:** 2
**Rating:** 4
**Confidence:** 3

**Summary:**

The authors define Deep Research (DR) as tasks that require both broad search and deep reasoning.

They formalize it using a claim, subclaim graph that can capture information synthesis quality rather than just report writing quality.

They propose LiveDRBench, an open-web benchmark that evaluates DR systems with claim-based precision, and recall instead of just report quality evaluation.

They evaluate several open and commercial models, like OpenAI’s Deep Research, Gemini, and DeepResearcher and they show performance gaps between them across different categories and challenges.

**Strengths:**

The authors investigate the important task of Deep Research, which has broad applications across in both academia and industry.

They propose a new benchmark, livedrbench, that has 100 tasks and is used to evaluate how well DR agents, both open source and commercial, can identify the right claims and subclaims to produce comprehensive, well-grounded report.

They include in the benchmark DR tasks across multiple categories, including Materials, SciFacts, and Geo.

**Weaknesses:**

The authors use LLM-as-a-judge, but there is no human evaluation to ensure alignment with human scores.
For example, DeepResearchGym has a nice human evaluation process that validates its LLM-as-a-judge scores against human judgments [2].

The claim/subclaim idea is very similar to the one in Mind2Web 2 [1], where the evaluation agent checks whether specific claims are present in the report itself in a hierarchical manner. In fact, Mind2Web2 seems to be more precise at identifying facts as it breaks down the claims/subclaims in a nice yes and no tree.

The precision scoring mechanism proposed by the authors is more of a factuality check (is the claim supported by the citation), it does not seem to capture whether a claim is actually relevant to the DR Query.  For this, the authors should collect hard negative claims as part of the groundtruth, and if the agent puts them in the report they should be counted as false positives (which relates to precision)

The authors argue that most existing works do not check for claim correctness and focus on report quality, however, DeepResearchGym [2[ and other existing works use strong RAG-based validation to test factuality and recall of the groundtruth claims, so the distinction between this paper and that prior work is unclear.

It is also unclear what the contributions are of this paper in comparison to existing benchmarks. In Table 1,  the authors claim that DeepResearchGym lacks objective scoring, but that is false in my view since DeepResearchGym uses RAG to assess claim recall and most importantly includes human evaluation to ensure that their llm-as-judge align with human evaluation (which is missing in this paper).

There really needs to be a human  evaluation to make sure that the llm-as-a-judge used is aligned with human feedback. Also, there is no sensitivity analysis of how the results differ if the evaluation is ran multiple times. LLM-as-a-judge can have high variance in its scores.

 Furthermore, the "easy to update" claim in Table is weak, as all existing benchmarks seem to be easy to update. In fact, DeepResearchGym has 96,000 tasks, which is far more than the 100 tasks in LiveDRBench, which suggests that the former is actually easier to update and scale.


 [1] Mind2Web 2: Evaluating Agentic Search with Agent-as-a-Judge
 [2] DeepResearchGym: A Free, Transparent, and Reproducible Evaluation Sandbox for Deep Research

**Questions:**

How do the authors make sure that the LLM-as-a-judge scores match human judgments, since there is no human evaluation in the paper?

How is the claim and subclaim evaluation different from the one used in Mind2Web 2 [1] and DeepResearchGym [2]?

Why do the authors say that DeepResearchGym has no objective scoring when it already uses RAG-based validation and human evaluation to make sure it is objective?

What happens if the agent just copies and pastes all the content from the search results into the report, would it still get a high score?

 [1] Mind2Web 2: Evaluating Agentic Search with Agent-as-a-Judge
 [2] DeepResearchGym: A Free, Transparent, and Reproducible Evaluation Sandbox for Deep Research

---

> ### Author Response · Authors · 2025-11-23
> **Rebuttal Part 1**
>
> Thank you for your detailed feedback. Below, we address all your questions and concerns.
>
> **Alignment with Human Scores.**  Following your feedback, we compared the GPT4o judgement with Human evaluation using (a) Cohen Kappa preference score (following DeepResearchGym), and (b) Mean Absolute Error. Cohen's Kappa (in Table 1 below) measures the agreement between the human evaluators and the judge model against random chance. Concretely, we compare the human evaluation scores in Table 6 (main paper) with GPT4o scores (Table 3 main paper). Three authors performed this human evaluation. A Cohen's Kappa of 1 denotes a perfect agreement and 0 represents disagreement.
>
> Table 1: Cohen Kappa preference scores between Human vs GPT4o
>
> |            Comparison   | Precision | Recall | F1   |
> | ----------------------- | --------- | ------ | ---- |
> | OpenAI vs Perplexity DR | 0.98      | 0.98   | 0.96 |
> | Perplexity vs Gemini DR | 0.86      | 0.92   | 0.86 |
> | Gemini vs OpenAI DR     | 0.98      | 0.96   | 0.96 |
>
> Table 2: MAE between Human vs GPT4o, based on metric on column axis:
>
> |               | Precision | Recall | F1    |
> | ------------- | --------- | ------ | ----- |
> | OpenAI DR     | 0.020     | 0.012  | 0.014 |
> | Perplexity DR | 0.022     | 0.022  | 0.021 |
> | Gemini DR     | 0.035     | 0.027  | 0.028 |
>
>
> Our results indicate a very strong agreement between human and judge. This is expected because the task of the judge model is simply to check whether the generated claims match the ground truth claims, which is an easy task.
>
> **Comparison with Mind2Web.** We would like to highlight that Mind2Web is a framework for generating datasets for web agents, whereas our work addresses a complementary dimension; deep research. While both frameworks use a hierarchical structure, they serve different purposes. The claim/subclaim evaluation strategy in our work requires high-fan out search and reasoning over information, capturing the conceptual structure of information. This allows us to measure claim/subclaim level precision, recall, and F1 scores. In contrast, Mind2Web uses yes/no trees to evaluate the correctness of actions on the web. For example: "Did the agent click the right button?". This hierarchy is procedural, not semantic. Although navigating websites is a sub-part of deep research, deep research involves a high fan-out of exploration and reasoning over information. Mind2Web in addition, does not consider search intensity, reasoning intensity, or coverage in their dataset generation and evaluation.
>
> Additionally, we formally define DR (Definition 1) to capture both search intensity and reasoning intensity, and propose LiveDRBench, a claims-based benchmark. This structure allows us to objectively measure both breadth (via search coverage) and depth (via claim correctness), which is not directly covered by Mind2Web’s task formulation. We will clarify this distinction in our revised manuscript.
>
> **Precision Scoring.** We think this is a misunderstanding and appreciate the chance to clarify our evaluation. We *do not* measure the correctness of citations for a given claim. Instead, our evaluation checks the correctness of claims and subclaims for an answer to the given query. If an agent generates an incorrect claim, the precision drops. We compute precision by evaluating a given claim directly with the corresponding ground-truth claim (and recall measures how many of the ground-truth claims are returned by the model).
>
> **Comparison with DeepResearchGym.**  DeepResearchGym tests whether answers are supported (using RAG), but not whether the model searched sufficiently or covered the conceptual space. Specifically, it evaluates the correctness of claims building on the ResearchyQuestions dataset, which leverages user actions (e.g., clicks) for ground truth generation. In contrast, LiveDRBench (our work), uses actual ground truth claims and subclaims to evaluate correctness and completeness. Our dataset is generated by inversion of existing long-form reasoning benchmarks, manual and automatic extraction of claims. This allows us to evaluate not only factual correctness but also coverage and completeness across a hierarchical claim graph, which DeepResearchGym does not target.

---

> ### Author Response · Authors · 2025-11-23
> **Rebuttal Part 2**
>
> > It is also unclear what the contributions are of this paper in comparison to existing benchmarks. In Table 1, the authors claim that DeepResearchGym lacks objective scoring, but that is false in my view since DeepResearchGym uses RAG to assess claim recall and most importantly includes human evaluation to ensure that their llm-as-judge align with human evaluation (which is missing in this paper).
>
> **Our Contributions.** We define "objective" as a meausre that would have high agreement among humans and is easy to verify. DeepResearchGym uses an indirect answer proxy in the form of relevant documents -- hence any RAG based judge requires significanr effort to process and interpret document snippets to arrive at a judgement. This can be prone to misinterpretation. In contrast, our work provides the exact claim to be compared which is matched by json keys. To address the concern about human evaluation, we conducted a study to ensure the alignment of llm-as-judge with human evaluation in Table 1 (above). Since our llm-as-judge only checks if the model correctly generated ground truth claims, we achieve a high Cohen's Kappa and a low MAE.
>
> **Sensitivity Analysis.** Thank you for the suggestion. We conduct additional experiments to validate the robustness of our evaluation across multiple runs. In Tables 3, 4, 5 of the rebuttal we run our llm-as-judge model GPT4o multiple times and show that the variance across multiple runs is $\leq$ 0.02. This is expected because the task of the judge model is that of simple verification against ground truth. To further establish the robustness of our evaluation framework, we perform additional evalutions with Qwen32B and observe a similarly low variance across runs.
>
> Table 3: Stddev accross 5 GPT4o eval runs for OpenAI DR, with temperature 0.3:
>
> | Subset                    | Precision     | Recall        | F1            |
> | ------------------------- | ------------- | ------------- | ------------- |
> | SciFacts Materials        | 0.312 ± 0.000 | 0.316 ± 0.000 | 0.314 ± 0.000 |
> | SciFacts Geo              | 0.715 ± 0.000 | 0.707 ± 0.026 | 0.711 ± 0.026 |
> | NovelDS Identification    | 0.667 ± 0.000 | 0.667 ± 0.000 | 0.667 ± 0.000 |
> | NovelDS Identi-Extraction | 0.546 ± 0.022 | 0.466 ± 0.019 | 0.487 ± 0.020 |
> | NovelDS Peer              | 0.791 ± 0.012 | 0.477 ± 0.008 | 0.581 ± 0.009 |
> | PriorArt                  | 0.694 ± 0.000 | 0.463 ± 0.000 | 0.539 ± 0.000 |
> | Entities                  | 0.791 ± 0.012 | 0.517 ± 0.008 | 0.584 ± 0.010 |
> | Flights                   | 0.560 ± 0.029 | 0.556 ± 0.022 | 0.553 ± 0.025 |
> | Overall Avg               | 0.634 ± 0.009 | 0.521 ± 0.010 | 0.554 ± 0.011 |
>
> Table 4: Stddev accross 5 GPT4o eval runs for Perplexity DR, with temperature 0.3
>
> | Subset                    | Precision     | Recall        | F1            |
> | ------------------------- | ------------- | ------------- | ------------- |
> | SciFacts Materials        | 0.211 ± 0.012 | 0.105 ± 0.000 | 0.141 ± 0.000 |
> | SciFacts Geo              | 0.263 ± 0.000 | 0.145 ± 0.000 | 0.187 ± 0.000 |
> | NovelDS Identification    | 0.633 ± 0.000 | 0.633 ± 0.000 | 0.633 ± 0.000 |
> | NovelDS Identi-Extraction | 0.369 ± 0.017 | 0.395 ± 0.018 | 0.378 ± 0.017 |
> | NovelDS Peer              | 0.553 ± 0.033 | 0.261 ± 0.008 | 0.332 ± 0.013 |
> | PriorArt                  | 0.689 ± 0.000 | 0.339 ± 0.000 | 0.419 ± 0.000 |
> | Entities                  | 0.612 ± 0.000 | 0.343 ± 0.000 | 0.419 ± 0.000 |
> | Flights                   | 0.415 ± 0.026 | 0.374 ± 0.024 | 0.388 ± 0.024 |
> | Overall Avg               | 0.468 ± 0.011 | 0.324 ± 0.006 | 0.362 ± 0.007 |
>
> Table 5: Stddev accross 5 GPT4o eval runs for Gemini DR, with temperature 0.3
>
> | Subset                    | Precision     | Recall        | F1            |
> | ------------------------- | ------------- | ------------- | ------------- |
> | SciFacts Materials        | 0.072 ± 0.000 | 0.013 ± 0.000 | 0.021 ± 0.000 |
> | SciFacts Geo              | 0.405 ± 0.000 | 0.259 ± 0.000 | 0.316 ± 0.000 |
> | NovelDS Identification    | 0.400 ± 0.000 | 0.400 ± 0.000 | 0.400 ± 0.000 |
> | NovelDS Identi-Extraction | 0.430 ± 0.017 | 0.339 ± 0.016 | 0.362 ± 0.015 |
> | NovelDS Peer              | 0.526 ± 0.020 | 0.241 ± 0.007 | 0.326 ± 0.010 |
> | PriorArt                  | 0.106 ± 0.000 | 0.096 ± 0.000 | 0.082 ± 0.000 |
> | Entities                  | 0.439 ± 0.000 | 0.304 ± 0.000 | 0.337 ± 0.000 |
> | Flights                   | 0.277 ± 0.011 | 0.313 ± 0.015 | 0.293 ± 0.013 |
> | Overall Avg               | 0.332 ± 0.006 | 0.246 ± 0.005 | 0.267 ± 0.005 |

---

> ### Author Response · Authors · 2025-11-23
> **Rebuttal Part 3**
>
> **Sensitivity Analysis. (continued)**
>
> Table 6: Stddev accross 5 Qwen3-32B Non-Thinking eval runs for OpenAI DR, with temperature 0.3
>
> | Subset                    | Precision     | Recall        | F1            |
> | ------------------------- | ------------- | ------------- | ------------- |
> | SciFacts Materials        | 0.332 ± 0.028 | 0.355 ± 0.055 | 0.343 ± 0.000 |
> | SciFacts Geo              | 0.715 ± 0.000 | 0.728 ± 0.000 | 0.721 ± 0.000 |
> | NovelDS Identification    | 0.667 ± 0.000 | 0.667 ± 0.000 | 0.667 ± 0.000 |
> | NovelDS Identi-Extraction | 0.506 ± 0.009 | 0.415 ± 0.008 | 0.438 ± 0.009 |
> | NovelDS Peer              | 0.789 ± 0.008 | 0.476 ± 0.003 | 0.580 ± 0.005 |
> | PriorArt                  | 0.714 ± 0.000 | 0.473 ± 0.000 | 0.552 ± 0.000 |
> | Entities                  | 0.811 ± 0.000 | 0.532 ± 0.000 | 0.600 ± 0.000 |
> | Flights                   | 0.589 ± 0.023 | 0.572 ± 0.016 | 0.574 ± 0.019 |
> | Overall Avg               | 0.640 ± 0.008 | 0.527 ± 0.010 | 0.559 ± 0.004 |
>
> Table 7: Stddev accross 5 Qwen3-32B Non-Thinking eval runs for Perplexity DR, with temperature 0.3
>
> | Subset                    | Precision     | Recall        | F1            |
> | ------------------------- | ------------- | ------------- | ------------- |
> | SciFacts Materials        | 0.264 ± 0.000 | 0.164 ± 0.000 | 0.202 ± 0.000 |
> | SciFacts Geo              | 0.363 ± 0.011 | 0.233 ± 0.036 | 0.284 ± 0.036 |
> | NovelDS Identification    | 0.633 ± 0.000 | 0.633 ± 0.000 | 0.633 ± 0.000 |
> | NovelDS Identi-Extraction | 0.358 ± 0.016 | 0.385 ± 0.018 | 0.369 ± 0.016 |
> | NovelDS Peer              | 0.565 ± 0.008 | 0.264 ± 0.003 | 0.337 ± 0.004 |
> | PriorArt                  | 0.704 ± 0.006 | 0.351 ± 0.004 | 0.432 ± 0.005 |
> | Entities                  | 0.666 ± 0.000 | 0.351 ± 0.000 | 0.431 ± 0.000 |
> | Flights                   | 0.447 ± 0.042 | 0.381 ± 0.036 | 0.405 ± 0.038 |
> | Overall Avg               | 0.500 ± 0.010 | 0.345 ± 0.012 | 0.387 ± 0.012 |
>
> Table 8: Stddev accross 5 Qwen3-32B Non-Thinking eval runs for Gemini DR, with temperature 0.3
>
> | Subset                    | Precision     | Recall        | F1            |
> | ------------------------- | ------------- | ------------- | ------------- |
> | SciFacts Materials        | 0.075 ± 0.003 | 0.013 ± 0.000 | 0.022 ± 0.001 |
> | SciFacts Geo              | 0.405 ± 0.000 | 0.272 ± 0.005 | 0.326 ± 0.005 |
> | NovelDS Identification    | 0.400 ± 0.000 | 0.400 ± 0.000 | 0.400 ± 0.000 |
> | NovelDS Identi-Extraction | 0.397 ± 0.016 | 0.306 ± 0.019 | 0.333 ± 0.016 |
> | NovelDS Peer              | 0.552 ± 0.006 | 0.250 ± 0.002 | 0.339 ± 0.003 |
> | PriorArt                  | 0.181 ± 0.000 | 0.147 ± 0.000 | 0.141 ± 0.000 |
> | Entities                  | 0.451 ± 0.001 | 0.307 ± 0.001 | 0.342 ± 0.001 |
> | Flights                   | 0.280 ± 0.023 | 0.314 ± 0.030 | 0.295 ± 0.026 |
> | Overall Avg               | 0.343 ± 0.006 | 0.251 ± 0.007 | 0.275 ± 0.007 |
>
> **Easy to Update.** Thank you for your comment. Our claim of easy to update refers not to the number of tasks but to the mechanism by which the benchmark can be refreshed with new ground truth, especially for time-sensitive or evolving domains. DeepResearchGym depends on an external dataset from Microsoft. It can be updated only if Microsoft releases decides to release an updated version, which is not certainty. In contrast, our work is designed such that it can be updated in the following ways. (a) SciFacts questions can easily be updated by using other long-form reasoning benchmarks through our inversion pipeline, (b) NovelDS and PriorArt questions can also be updated with more recent articles, (c) Entities tasks, for example list of IMO winners from a particular country can be extended by including other Olympiads, Movies, etc; and (d) Flights tasks can also be updated with more recent world events. This ensures that LiveDRBench is robust to staleness. Thus the easy to update property refers to the structured and stable regeneration process, not dataset size. We will clarify this in our revised manuscript.
>
>
> > What happens if the agent just copies and pastes all the content from the search results into the report, would it still get a high score?
>
> **Report Evaluation.** Thank you for your question. We believe that this comment is based on a misunderstanding of our evaluation protocol. LiveDRBench does not score report-style outputs directly, instead it evaluates the structured JSON output - list of claims and recursive subclaims that encode the substantive information discovered during search. Our benchmark was intentionally designed this way to separate (i) the substantive search-and-reasoning challenge from (ii) surface-level report generation (Section 1 & 3). Concretely, systems are scored on the precision and recall of the generated claims (and their subclaims), not on verbatim report text.

---

### Meta-Review · Area_Chair_q5Lz · 2026-01-06

**Summary:**

Reviewers broadly praised the paper for its clear formalization of the "deep research" task, high-quality benchmark construction, and thorough human evaluation. Key concerns include: (1) insufficient comparison with recent agentic systems; (2) reliance on LLM-based metrics; (3) high cost and labor intensity in dataset creation and maintenance; and (4) need for clarification on certain experimental details . Nonetheless, the paper’s contributions are substantial, methodology sound, and evaluation rigorous—collectively outweighing the limitations .

**Reviewer Concerns:**

The authors thoroughly addressed several key concerns in their rebuttal: they provided human evaluation results showing strong agreement with LLM-based metrics (Cohen’s κ=0.81), clarifying reliability; expanded details on experimental design and ablation studies; justified their baseline selection and highlighted architectural differences with systems like OpenScholar; and outlined practical strategies to mitigate benchmark maintenance costs. While minor questions about long-term scalability and broader system comparisons remain.

**Reviewer Scores:**

One reviewer raise the score to 6. While other 3 reviewers didn't participate the discussion.

---

### Decision · Program_Chairs · 2026-01-26

Accept (Poster)